# CONTENT-STYLE DISENTANGLED REPRESENTATION FOR CONTROLLABLE ARTISTIC IMAGE STYLIZATION AND GENERATION

## ABSTRACT

Controllable artistic image stylization and generation aims to render the content provided by text or image with the learned artistic style, where content and style decoupling is the key to achieve satisfactory results. However, current methods for content and style disentanglement primarily rely on image information for supervision, which leads to two problems: 1) models can only support one modality for style or content input;2) incomplete disentanglement resulting in semantic interference from the reference image. To address the above issues, this paper proposes a content-style representation disentangling method for controllable artistic image stylization and generation. We construct a *WikiStyle+* dataset consists of artworks with corresponding textual descriptions for style and content. Based on the multi-modal dataset, we propose a disentangled content and style representations guided diffusion model. The disentangled representations are first learned by Q-Formers and then injected into a pre-trained diffusion model using learnable multi-step cross-attention layers for better controllable stylization. This approach allows model to accommodate inputs from different modalities. Experimental results show that our method achieves a thorough disentanglement of content and style in reference images under multimodal supervision, thereby enabling a harmonious integration of content and style in the generated outputs, successfully producing style-consistent and expressive stylized images. The code of our method will be available upon acceptance.

## 1 INTRODUCTION

Artistic image stylization and generation task aims at creating new images by applying specific artistic styles to content. In stylization, the concept of "content" is well-defined, typically referring to the subject of an image or the semantics of input text. However, the definition of "style" is relatively vague and lacks consistent standards. Art historian, Meyer Schapiro, has defined artistic style as: "The constant form and sometimes the constant elements, qualities, and expression in the art of an individual or a group" (Karkov & Brown, 2003). For example, Impressionism emphasizes the natural representation of light and color. Therefore, to truly capture and reproduce artistic style, stylization models must learn the unique ways of artists handle and present content in their creative process. To achieve this, the model needs to effectively disentangle and control content and style during the stylization process, thereby producing visually coherent and stylized results.

Recently, diffusion models (Ho et al., 2020; Rombach et al., 2022) have demonstrated great potential in text-to-image stylization tasks (Ramesh et al., 2022; Ye et al., 2023; Mou et al., 2024; Chen et al., 2024a) due to their powerful generative capabilities. These methods typically extract reliable features in the reference images serving as conditional information to guide the diffusion model to follow the predetermined style. However, the features extracted by the encoder often couple style and semantics, leading to semantic conflicts between the text and the reference image . This in turn causes content leakage from the reference image, as shown in Fig. 1 (a). Some approaches (Xing et al., 2024) (Qi et al., 2024) attempt to achieve disentanglement by using separate encoders to extract style and content representations. Still, they encounter the following issues: 1) These methods depend on datasets consisting of paired content and stylized images for training. Besides, the disentanglement process is solely reliant on image-based supervision, which limits the style input to only image

modality. 2) They primarily focus on extracting style representations from the reference image but do not effectively decouple the content and style. This results in a mismatch between style and content in the generated images, failing to achieve the desired stylization effect (as shown in Fig.1 1(b)). Additionally, they can only handle visual elements from single image and unable to mimic the style of an artist or genre. Artbank (Zhang et al., 2024) achieves artist-based stylization by constructing a style prompt bank to store knowledge from art collections, which needs to be retrained when extend to new artists. However, Artbank is trained with the reconstruction objective with image-based supervision, which leads to unsatisfied performance in text-controlled image generation.

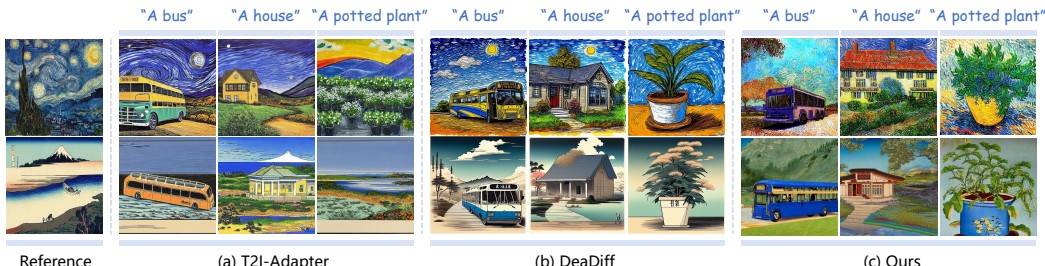

Figure 1: Given a style reference image, our model is capable of synthesizing controllable artistic images that resemble the style and faithful to text contents simultaneously. However, previous encoder-based methods like T2I-Adapter Mou et al. (2024) shows content leakage while previous autoencoder-based methods like DeaDiff Qi et al. (2024) fails to achieve the desired stylization.

To address the aforementioned issues, this paper proposes a content-style representation disentanglement method for controllable artistic image stylization and generation. First, we constructed a multimodal art image-text dataset, WikiStyle+. We collected art images and their corresponding style information from the WikiArt website, including details such as artists, creation periods, genres, and painting medias, as references for style descriptions. Additionally, we used large language model to generate textual descriptions of the content of these artworks. In this way, we tackled the problem of the lack of explicitly disentangled data for style and content in artistic images from a multimodal perspective. Based on the constructed WikiStyle+ dataset, we proposed a disentangled content and style representations guided diffusion model. Through multimodal alignment tasks, the Q-former aligns the learned image style features with style descriptions and the learned content features with content descriptions. This method utilizes multimodal data to provide the style and content information of the reference images for disentanglement supervision, achieving explicit separation of content and style. As a result, the model learns style representations consistent with the definition of style in art history, which reflects the distinct ways artists depicts the world. Building on this, we inject the learned style and content representations into the multi-step cross-attention layers of the diffusion model, enabling text-to-image stylization and generation.

With the explicit disentanglement of style and content, our method can adjust visual elements based on content and different style prompts, making stylization go beyond mere visual imitation and achieve a harmonious integration of content and style in the generated results, as shown in Fig. 1 (c). Our approach can effectively prevent content leakage from reference image. Besides, the multimodal disentangle supervision allows us to accommodate style and inputs from either image or text modalities flexibly, which is typically restricted by previous image-supervised decoupling methods (e.g., DeaDiff Qi et al. (2024)). In summary, our contributions are threefold:

- We constructed the art image-text dataset, WikiStyle+, addressing the problem of the lack of explicitly disentangled data for style and content from a multimodal perspective.

- We propose a disentanglement method that explicitly decouples style and content representations through multimodal supervision, enabling the model to accept inputs from different modalities as control conditions.

- We constructed a disentangled representation-guided diffusion model, where the disentangled content and style representations are injected into the cross-attention layers at different time steps of the diffusion model, successfully generating images with harmonious integration of content and style.

## 2 RELATED WORK

### 2.1 DIFFUSION-BASED IMAGE STYLIZATION WITH MULTIMODAL LATENTS

In recent years, Diffusion Probabilistic Models Sohl-Dickstein et al. (2015) have demonstrated immense potential in the field of image generation Dhariwal & Nichol (2021); Ho et al. (2020); Song et al. (2020).Thanks to large-scale multimodal pre-trained models Radford et al. (2021); Li et al. (2023), diffusion models have achieved remarkable success in text-to-image generation tasks Ramesh et al. (2022); Rombach et al. (2022); Saharia et al. (2022). Text-to-image stylization tasks based on diffusion models are primarily divided into two categories: one is optimization-based methods, which involves fine-tuning the diffusion model to enable it to generate images for a specific style Ruiz et al. (2023); Kumari et al. (2023); Sohn et al. (2023), or utilizing textual inversion to redefine or optimize textual embeddings related to a specific style Gal et al. (2022); Zhang et al. (2023), thereby generating more accurately stylized images that match with descriptions. The other category involves conditional diffusion models based on pre-trained encodersHuang et al. (2023); Li et al. (2024); Mou et al. (2024); Wang et al. (2023b); Ye et al. (2023); Zhao et al. (2024), which extract stylistic features from reference images using pre-trained encoders and then inject these features as conditions into diffusion model to generate images with specific styles.

### 2.2 CONTENT-STYLE DECOUPLING

In image stylization tasks, decoupling content and style is key to enhancing the quality of generated images. Text inversion methods (such as InST (Zhang et al., 2023), VCT (Cheng et al., 2023), DreamBooth (Ruiz et al., 2023), Textual Inversion (Gal et al., 2022), Custom Diffusion (Kumari et al., 2023), and ArtBank (Zhang et al., 2024)) map reference images the embedding space of special text tokens through a reversal module. They construct text prompts to specify whether the special tokens represent content or style, thereby separately decoupling content and style, such as "a dog in [v] style" or "a [v] dog in Van Gogh style," where [v] represents either style or content information.

For unoptimized attention mechanism methods like StyleAligned (Hertz et al., 2024) and Visual Style Prompting (Jeong et al., 2024) typically achieve zero-shot content-style decoupling for image stylization by sharing or optimizing attention mechanisms in SD. Cross-domain alignment methods like StyleDiffusion (Wang et al., 2023a) and OSASIS (Cho et al., 2024) use cross-domain and intra-domain losses in CLIP space to achieve latent content-style decoupling.

Adapter-based methods (such as IP-Adapter (Ye et al., 2023), T2I-Adapter (Mou et al., 2024), and StyleAdapter (Wang et al., 2023b)) adjust specific layers or channels of the model, allowing it to effectively separate and blend style and content without altering its original architecture. Adapter methods based on pre-trained encoders (such as InstantStyle (Wang et al., 2024) and DEA-Diffusion (Qi et al., 2024)) decouple the content and style of reference images in the encoding space. However, current methods fail to achieve a thorough decoupling of content and style from an artistic creation perspective, resulting in content leakage from the reference image in the generated results.

## 3 DATASET

### 3.1 DATASET CONSTRUCT PIPELINE

A significant challenge in content-style decoupling is the lack of paired dataset. Current methods construct content-stylized image pairs using proposed prompts with pretrained text-to-image generation models. However, due to the randomness of generative models, even when using the same prompt, there are significant differences between the style and content of the images generated, necessitating tedious manual data cleaning. Fortunately, existing large language models have demonstrated a strong fidelity in generating image descriptions. We can construct a content description-style description-artwork triplet dataset to addressing the problem of the lack of explicitly disentangled data for style and content from the multimodal perspective. Moreover, the impact of randomness in text generation on semantics is far less than the impact of randomness in image generation on style.

Formally, the construction of the paired datasets involves the following three steps:

**Step 1. Art image collection.** We collect 189,631 data entries from the WikiArt websiteWikipedia (2021). The gathered data includes artistic images, along with detailed professional information provided by WikiArt for each artwork. This information pertains style-related attributes like the author, style, genre, media, and other relevant labels.

**Step 2. Art image selection.**Upon analyzing the data, we observe that works like photography, architecture, design drawings, advertisements, and illustrations often lack distinct stylistic features, while art movements like abstract and minimalism usually lack specific subjects. Based on these findings, we filter out works labeled under these genres, ultimately forming an art dataset comprising 146,547 records.

WikiStyle+

**Content Description**
A bridge over a river with people on horseback crossing the bridge and a boat with passengers on the river.

**Style Description**
Artist is Vasily Sadovnikov, date is 1830, style is Realism, genre is cityscape.

**Content Description**
A garden with a palm tree and a house in the background.

**Style Description**
Artist is Lili Elbe, date is 1918, style is Post-Impressionism, genre is landscape.

Figure 2: Examples from WikiStyle+ dataset, each item contains artwork, content text and style text.

**Step 3. Content description generation.**To construct paired content and style data, we utilized the recently open-sourced InternVL-Chat Chen et al. (2024b) with the prompt, "<image>, describe the content of this picture briefly." to generate content descriptions for each artwork. This approach enabled us to obtain both the content and style information of the artistic images from a multi-modal perspective, effectively addressing the issue of insufficient explicit data for content-style disentanglement.

## 3.2 DATASET DETAILS

The final dataset comprises 146,547 triplets of art images, style descriptions, and content descriptions. Based on the professional annotation on WikiArt, we selected 4 style attributes under different categories, including artist (2,789), artistic style (209), genre (63), and mediums (184). In terms of content, it covers a variety of themes such as portraits, still lifes, natural landscapes, and cultural landscapes. More details about WikiStyle+ are provided in Appendix A.1.

## 4 METHOD

We propose a disentangled content and style guided diffusion model for controllable artistic image stylization and generation, as shown in Fig. 3. In Sec. 4.1, we present our proposed Content and Style Disentangled Network (CSDN) based on a pre-trained autoencoder van den Oord et al. (2018), it outputs disentangled content and style representations for the followup diffusion model. In Sec. 4.2, we introduce the Multi-step Cross-attention Layers (MCL) for controllable artistic image stylization and generation by injecting the disentangled representations into a pre-trained Stable Diffusion (SD) Rombach et al. (2022) model.

## 4.1 CONTENT AND STYLE DISENTANGLEMENT NETWORK

The core of CSDN lies in the disentangled representation learning, where we employ Q-Former Li et al. (2023) to separate the style and content from images and align the feature spaces of images and text accordingly. Different from Qi et al. (2024) that uses two independent Q-Formers, we adopt a simpler design with two sets of learnable query embeddings, one dedicated to extracting content embeddings from multimodal inputs and the other for extracting style embeddings. There are two advantages by doing so: 1) Smaller model and faster convergence rate; 2) Physical decoupled query embeddings are crucial for explicit disentanglement of content and style.

The dataset is structured as triplets (image, style text, content text) as introduced in Sec. 3, which gives us data basis for disentangling content and style altogether. The style text is created by concatenating fields related to style in the dataset, such as artist, date, style and so on, in the format of "<key >is <value >". The content text is derived from the description field in the dataset, generated by a multimodal large language model. We achieve content and style disentanglement by minimizing an objective function that incorporates the Image-Text Contrastive Learning Loss $\mathcal{L}_{itc}$, Image-Text

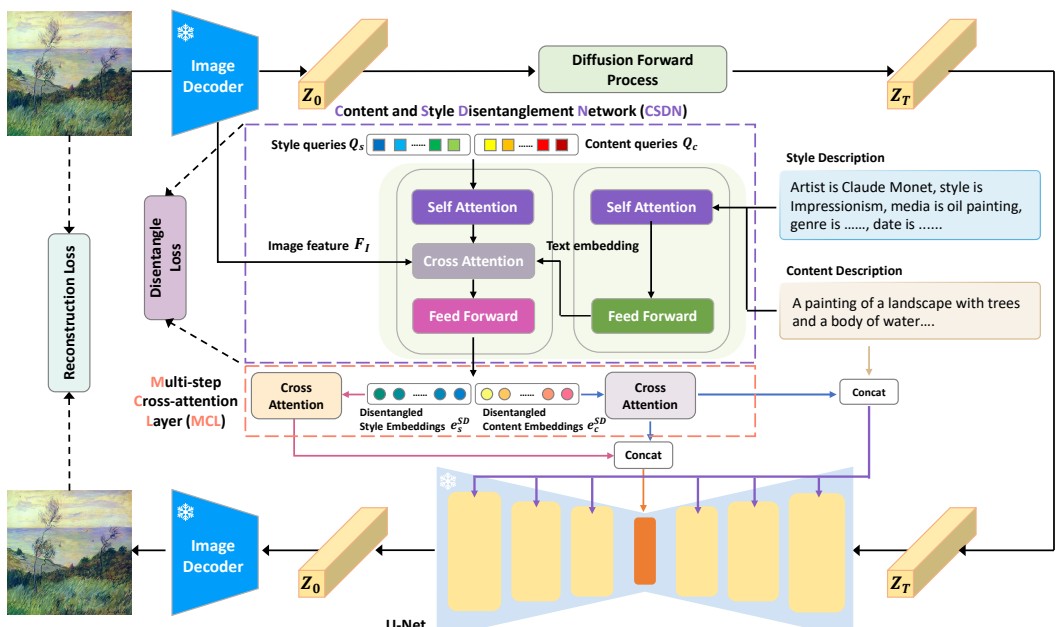

Figure 3: Overview of our model that contains three parts: 1) a pre-trained image encoder; 2) a Content and Style Disentangled Network (CSDN) with a connection to a pre-trained Stable Diffusion (SD) model; 3) a learnable multi-step cross-attention layers (MCL) to separately inject the content and style features into the SD model.

Matching Loss $\mathcal{L}_{itm}$, and Image-grounded Text Generation Loss $\mathcal{L}_{itg}$ as follows:

$$\mathcal{L}_s = \mathcal{L}_{itc}^s + \mathcal{L}_{itm}^s + \mathcal{L}_{itg}^s, \quad \mathcal{L}_c = \mathcal{L}_{itc}^c + \mathcal{L}_{itm}^c + \mathcal{L}_{itg}^c, \quad \mathcal{L} = \mathcal{L}_s + \mathcal{L}_c \tag{1}$$

where the superscripts $s$ and $c$ separately denote style and content, $\mathcal{L}_s$ is the total style loss and $\mathcal{L}_c$ is the total content loss. The overall objective corresponds to the disentangle loss in Fig. 3.

**Image-Text Contrastive Learning.** The image is processed by a pre-trained image encoder to obtain image features $F_I$. The queries $Q$ composed of style queries and content queries then utilize the Q-Former to extract the visual representations of style $I_s$ and content $I_c$ from $F_I$, as well as the textual representations of style $T_s$ and content $T_c$ from the style and content text, respectively. Since the queries contain multiple output embeddings, we apply a pooling operation to them. Finally, we align visual representations $I_s$ and $I_c$ with textual representations $T_s$ and $T_c$ respectively.

$$\mathcal{L}_{itc} = -\frac{1}{N} \sum_{n=1}^{N} \left( \log \frac{\exp\left(d(I_n, T_n)/\tau\right)}{\sum_{j=1}^{N} \exp\left(d(I_n, T_j)/\tau\right)} + \log \frac{\exp\left(d(T_n, I_n)/\tau\right)}{\sum_{j=1}^{N} \exp\left(d(T_n, I_j)/\tau\right)} \right) \tag{2}$$

where $d(\cdot, \cdot)$ denotes the cosine distance, $\tau$ is a temperature scaling parameter, $N$ is the batch size. ITC enables the model to disentangle style and content effectively by easuring that features corresponding to style and content are aligned with their respective textual descriptions.

**Image-Text Matching.** ITM operates as a binary classification task, predicting whether an image-text pair is a positive or negative match. This enables the model to focus on fine-grained correspondence between images and text. ITM computes the cosine similarity between image embedding and text embedding, then using a linear layer to map the cosine similarity into matching probability. ITM uses binary classification loss to optimize the Q-former and classifier:

$$\mathcal{L}_{itm} = -\frac{1}{N} \sum_{n=1}^{N} [y_n \log P(y_n = 1|Pair_n) + (1 - y_n) \log P(y_n = 0|Pair_n)] \tag{3}$$

where $Pair_n$ represents the $n$-th image-text pair, $y_n$ is the ground truth label indicating whether the i-th image-text pair is a match ($y_n = 1$) or not ($y_n = 0$), $P(y_n = 1|Pair_n)$ is the model's predicted probability that the image-text pair is a match, $N$ is the batch size.

**Image-grounded Text Generation** trains the model to generate coherent style and content descriptions for given input image by predicting the next word based on the extracted embeddings $I_s$ and $I_c$ from the image using queries $Q$. A lightweight text decoder is used to generation the text sequence. It consists of two main components: a transformation module that applies a dense projection, an activation function, and layer normalization to refine hidden states, and a decoder layer that maps the processed hidden states to vocabulary logits using a linear layer.

$$P(w_m \mid w_1, w_2, \ldots, w_{m-1}, I) = \text{Decoder}(h_m) \tag{4}$$

where $h_m = f(w_1, w_2, \ldots, w_{m-1}, I)$ represents the hidden state at step $m$, generated based on the previous words and the extracted embeddings. At each step $m$, the decoder predicts the probability distribution for the next word, given the previously generated words and the extracted embeddings I.

ITG is implemented as the cross-entropy loss between the predicted probabilities and the groundtruth sequence:

$$\mathcal{L}_{itg} = -\sum_{m=1}^{M} \log P_\theta(w_m|I, w_{<m}) \tag{5}$$

where $P_\theta(w_m|I, w_{<m})$ is the probability of generating the next word $w_m$ given the image features $I$ and the preceding words $w_{<m}$, $M$ is the length of the text sequence. ITG encourages the model to learn robust textual representations of visual information, ensuring that the style and content embeddings extracted from an image is not only disentangled, but also interpretable and coherent.

## 4.2 ARTISTIC IMAGE GENERATIVE LEARNING STAGE

In the generative learning stage, We aim to feed the content and style embeddings of CSDN to a frozen SD model for controllable image generation. First, we project disentangled style and content representations $e_s$ and $e_c$ into the feature dimensions required by the SD model using a projection layer, resulting in the style embeddings and content embeddings for SD, $e_s^{SD}$ and $e_c^{SD}$, respectively. Then, we use multi-step learnable cross-attention layers (MCL) to inject the style and content embeddings into the denoising process of the SD model. At each timestep of the diffusion process, the style and content embeddings are introduced as conditions through the cross-attention layers in MCL to guide the generation process. These cross-attention layers embed the style and content embeddings into the current diffusion features using the attention mechanism:

$$Q = W_Q Z \tag{6}$$

$$K = \begin{cases} e_c^{SD} W_c^K, \\ Concat(e_c^{SD} W_c^K, e_s^{SD} W_s^K), & \text{for middle block of U-Net} \end{cases} \tag{7}$$

$$V = \begin{cases} e_c^{SD} W_c^V, \\ Concat(e_c^{SD} W_c^V, e_s^{SD} W_s^V), & \text{for middle block of U-Net} \end{cases} \tag{8}$$

$$Z_{new} = Softmax(\frac{QK^T}{\sqrt{d}})V \tag{9}$$

where $Z$ and $Z_{new}$ represent the noise states at the current step and the next step during the denoising process, respectively. Inspired by Wang et al. (2024), we inject the style embeddings only into the middle block of U-Net, which also benefits for preventing content leak. For the content text, we first extract text features using the original text encoder from SD, then concatenate the text features with the disentangled content representations $e_c$ extracted by CSDN, before injecting them into the diffusion process.

SD model initially transforms an input image $x$ into a latent code $z$. The noised latent code $z_t$ at timestep $t$ serves as the input for the denoising U-Net $\epsilon_\theta$ , which interacts with content prompts $c$ and style prompts $s$ through cross-attention. The supervision for this process is ensured by:

$$\mathcal{L}_{rec} = \mathbb{E}_{z,c,s,\epsilon \sim \mathcal{N}(0,1),t} \left[ \|\epsilon - \epsilon_\theta(z_t, t, c, s)\|_2^2 \right] \tag{10}$$

where $\epsilon \sim \mathcal{N}(\mathbf{0}, \mathbf{I})$ is a noise. The objective is corresponding to the reconstruction loss as in Fig. 3.

**Remark:** To prevent the model from becoming complacent during training by simply copying content from image features, which could diminish its generalization ability, we randomly replace the style or

content embeddings extracted from images with the corresponding embeddings extracted from texts. Both content and style embeddings are randomly selected from multiple modalities to participate in training the model. The logic behind this approach lies in feeding more complex tasks to train the model is beneficial for improving model's capabilities. This approach fosters a more nuanced and versatile understanding of both content and style, enabling the model to generate images that are both faithful to the input content and creatively infused with the desired style. Additionally, it also enables our model to generate outputs with various multimodal combinations. Finally, we randomly drop the keywords of style texts for enabling the model to accept a wider variety of style keyword combinations when using text as style prompts.

## 5 EXPERIMENT

### 5.1 EXPERIMENT SETTINGS

**Implementation Details.** We trained on L20-40G GPUs with 3750 total batches, using AdamW (Loshchilov, 2017) as the optimizer, with a learning rate of 5e-5, and performed 100 iterations. Regarding reasoning, for guidance without classifiers (Ho & Salimans, 2022), we use a scale of 7.5 and set T = 50 steps for DDIM (Song et al., 2020) sampling. All comparison methods were implemented using publicly available code and default settings. Detailed settings of our model are elaborated in Sec. A.2.

**Evaluation Metrics.** Based on previous works (Qi et al., 2024) (Jiang & Chen, 2024), the following metrics are used in the style transfer task. For content similarity, we use text alignment capability (TA), which calculate the cosine similarity in the CLIP embedding space between stylized images and their corresponding text prompts. For Image Quality (IQ), the LAION-Aesthetics Predictor V2 1 is adopted to evaluate the generation quality of the image. For style similarity (SS), we use a text prompt template to express style, namely "the painter is [v], the theme is [v]". Then, we measured the cosine similarity between the generated image and the corresponding style template in the CLIP text-image embedding space. We also conduct a user study to reflect the subjective preference (SP) for the results.

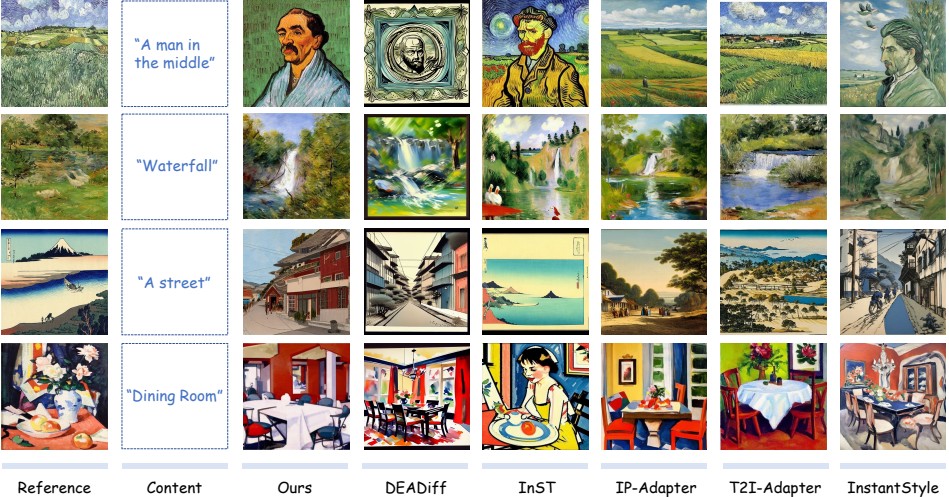

Figure 4: Qualitative comparison with the state-of-the-art text-to-image stylization methods.

### 5.2 COMPARISON WITH STATE-OF-THE-ARTS

In this section, we compare our method with the state-of-the-art methods. We introduce the experimental results based on supported input modalities and tasks as follows: 1) Text-to-image stylization, including optimization-based methods: InST (Zhang et al., 2023), encoder-based methods: IP-Adapter (Ye et al., 2023), DeaDiff (Qi et al., 2024), T2I-Adapter (Mou et al., 2024), and InstantStyle (Wang et al., 2024); 2) Text-to-image Generation, featuring advanced models

DallE (Ramesh et al., 2021) and Stable Diffusion (SD) (Rombach et al., 2022); 3) Collection-based stylization: Artbank (Zhang et al., 2024). More results please refer to Sec. A.3.

### 5.2.1 TEXT-TO-IMAGE STYLIZATION

Fig.4 illustrates the comparison results with state-of-the-art methods. For methods lacking effective decoupling mechanisms, such as IP-Adapter, T2I-Adapter and InST, semantic conflicts from the reference images are evident in the generated results, as shown in the first and third rows of Fig. 4. DEADiff and InstantStyle struggle to accurately mimic the style of the reference image when there is significant semantic gap between the reference image and the textual prompt. In contrast, our method not only faithfully follows textual prompts but also significantly mimics the artistic style contained in the reference images, making the stylized images closer to the true creations of the artists.

### 5.2.2 STYLIZED TEXT-TO-IMAGE GENERATION

Fig. 5 compares our method with advanced text-to-image generation methods. Due to the lack of a decoupling mechanism, Dall-E's generated images exhibit repetitive style patterns. Furthermore, the textures in the generated images do not resemble oil paintings. Our method, when prompted with different artists, generates distinct styles and adheres closely to the textual prompts, achieving performance comparable to that of Stable Diffusion. Due to our method's use of multimodal supervision during the pre-training phase, it also performs well in text-to-image generation tasks.

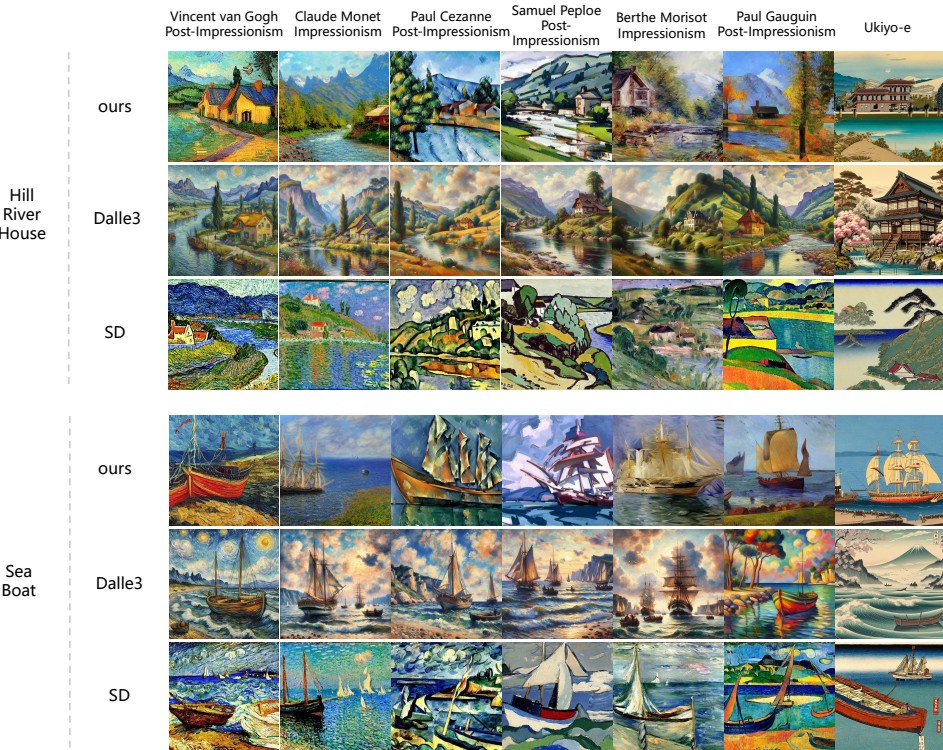

Figure 5: Qualitative comparison with the state-of-the-art text-to-image generation methods.

### 5.2.3 COLLECTION-BASED STYLIZATION

Fig. 6 presents a visual comparison between our method and ArtBank. Our approach shares a common feature with Artbank in that it learns the style information of art collections rather than the style of individual images. From Fig. 6, it is evident that our outputs more closely align with the artist's style and the content text. For instance, in first row, our results more closely resemble Van Gogh's portrait style compared to Artbank. In the stylized results with subject "barbecue", Artbank's outputs exhibit content leakage from the reference image, whereas our results effectively capture

Table 1: Quantitative comparison with the state-of-the-art text-to-image stylization methods.

| Metrics | InST | IP-Adapter | DEADiff | T2I-Adapter | InstantStyle | Ours |
|---|---|---|---|---|---|---|
| SS ↑ | 0.283 | 0.2882 | 0.236 | 0.276 | 0.280 | **0.293** |
| IQ ↑ | 5.845 | 5.856 | 5.891 | **5.895** | 5.798 | 5.811 |
| TA ↑ | 0.294 | 0.225 | 0.314 | 0.313 | 0.316 | **0.317** |
| SP ↑ | 2.333 | 2.167 | 2.583 | 2.250 | 3.333 | **3.857** |

the semantic of content prompt. Since Artbank also employs a reconstructive training approach, the image structure tends to be unclear when text is used as the content prompt, as demonstrated in the second and third rows of the images.

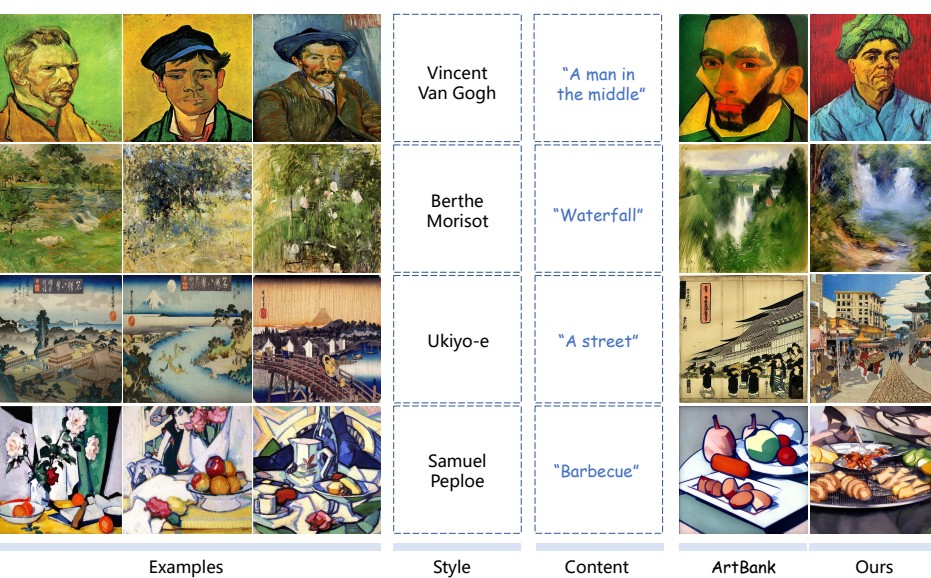

Figure 6: Qualitative comparison with the collection-based stylization methods.

### 5.2.4 QUANTITATIVE COMPARISON

Tab. 1 presents the style similarity, image quality, text alignment and the overall subjective preference of our method compared with the state-of-the-art methods. We can see that our method achieves the highest style similarity and content alignment, demonstrating that our method, through decoupling representations, effectively fixed the problem of semantic conflict and captured the overall artistic style. Furthermore, users demonstrate a significantly greater preference for our method over other ones. More detailed results and explanations could be found in appendix Sec.A.3.

### 5.3 CONTENT STYLE DECOUPLING

To verify the decoupling effect, we conducted an experiment using portrait paintings as content and style respectively. As shown in Fig. 7, the generated images depict the content themes accurately: middle-aged men (first two rows), a young boy (third row), and middle-aged women (fourth and fifth rows). Vertically, the styles of the reference images are also well-preserved, with distinct brushstrokes in the third and fourth columns. This demonstrates the robustness of the proposed content-style disentanglement method.

### 5.4 INPUT MODALITY ABLATION STUDY

To evaluate the effectiveness of multimodal representation training in the second phase, we conducted an ablation study. Table 2 shows the quantitative results under fixed combination and multimodal

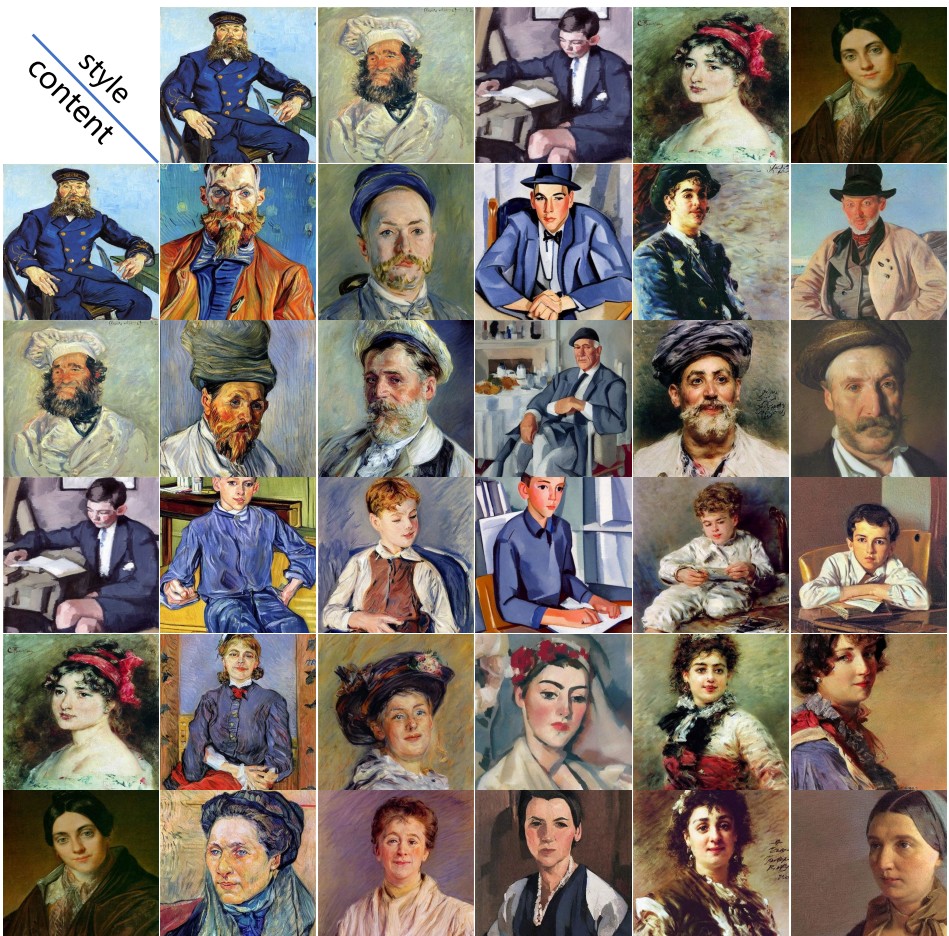

Figure 7: Qualitative results for content and style decoupling.

Table 2: Quantitative results for ablation study.

| Method | SS ↑ | IQ ↑ | TA ↑ |
|---|---|---|---|
| $T_c + I_s$ | 0.286 | 5.694 | 0.304 |
| multimodal | **0.293** | **5.811** | **0.308** |

settings. Training with only specific content-style combinations leads to a noticeable drop in metrics, while exposure to diverse features improves generalization across styles and visual characteristics. This demonstrates that mixed training with multimodal content and style features is crucial for generating artistically expressive images that align with both content and style.

## 6 CONCLUSION

In this paper, we proposed a method for disentangling content and style for artistic image stylization and generation. We constructed a multimodal art image-text dataset, *WikiStyle+*, to provide explicit data for supervised decoupling. We employed contrastive learning tasks to learn disentangled content and style representations, which then guided a diffusion model to generate stylized images. Our experiments across various tasks demonstrated the superiority of our method and highlighted the importance of effective content and style decoupling in image stylization.

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

## A APPENDIX

### A.1 DATASET DETAILS

The word cloud map shows the distribution statistics of the WikiStyle+ dataset in three main categories: artists (such as Van Gogh, Monet), genres (posters, mosaics), and media (oil paintings, puzzles). In Figure 8, we present the selected statistics of dataset and its word cloud. The average length of style descriptions is 19 words, while content descriptions average 48 words. Style descriptions are more concise, focusing on key elements such as artist and the influence of specific style movements. In contrast, content descriptions tend to be more detailed, providing an overview of the artwork as a whole and often including more specific elements.

### A.2 MODEL DETAILS

We implemented our method based on Stable Diffusion v1.5. We employ the ViT-L/14 from CLIP (Radford et al., 2021) as the image encoder, maintaining 16 queries to extract style representations and 16 queries to extract content representations. Based on the pre-trained weight provided by the BERT model (Brooks et al., 2023), we initialize the Q-Former and then fine-tune the parameters of Query and Q-Former on WikiStyle+. In the second stage, in order to adapt to Stable Diffusion (Rombach et al., 2022), we adopted an IP-adapter architecture and froze the parameters of UNet (Ronneberger et al., 2015) and the text encoder, concentrating the training work only on the style module and two projection layers, rather than training the entire model from scratch. This targeted approach ensures that we can effectively fine-tune the model for specific tasks while leveraging the

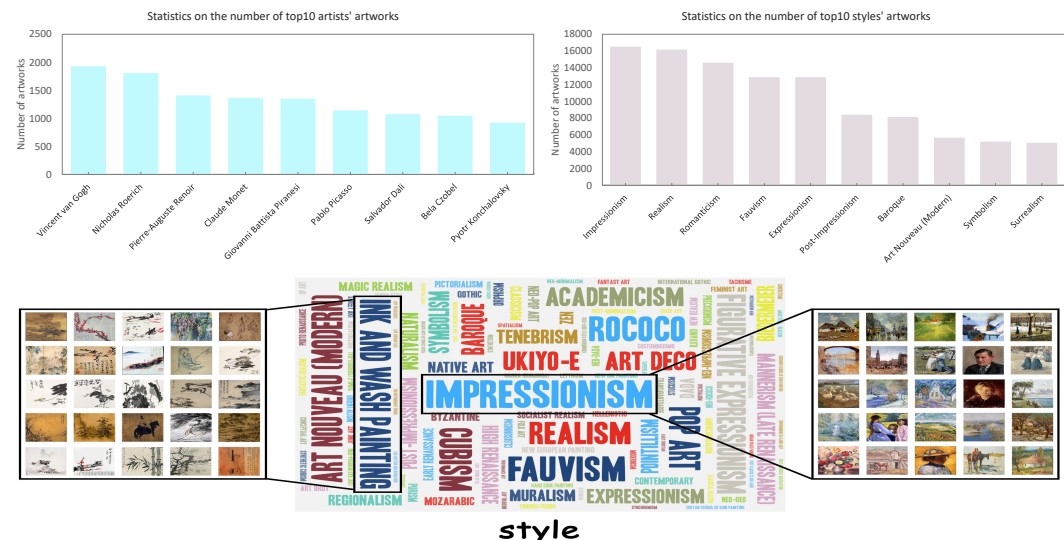

Figure 8: Some dataset details. **Top:** the selected statistics of our developed WikiStyle+ including top-10 artists and styles. **Bottom:** showcase of style word cloud.

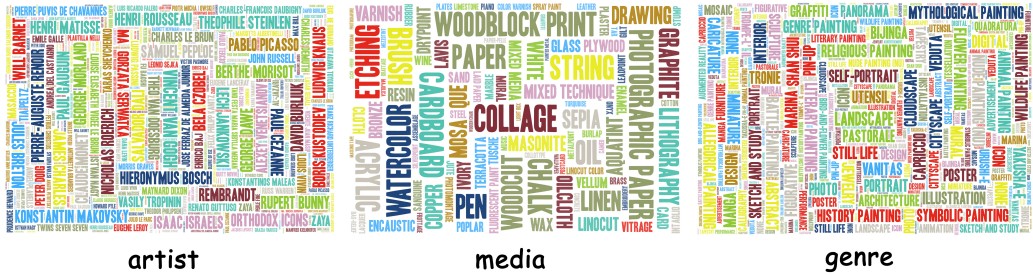

Figure 9: Word cloud for style attribute artist, media and genre.

powerful foundation provided by Stable Diffusion. In the second stage of training, we adopt three modal tasks for joint training to enhance model generalization. We set the sampling rates of the three tasks to 1:1:1, and jointly train the style module, projection layer, and Q-Former.

### A.3 SUPPLEMENTARY EXPERIMENTS

#### A.3.1 INFERENCE SPEED

We compared the inference speed of our method with two structurally similar methods, T2I-Adapter Mou et al. (2024) and Dea-Diff Qi et al. (2024) in Table 3. The complexity of our method is comparable to that of Dea-Diff, resulting in similar inference times.

| Method | T2I-Adapter Mou et al. (2024) | Dea-Diff Qi et al. (2024) | Ours |
|---|---|---|---|
| Inference Speed | 4.2s | 2.2s | 3.6s |

Table 3: Inference speed comparison.

#### A.3.2 QUANTITATIVE COMPARISON WITH SD AND ARTBANK

In Table 4, we present quantitative comparisons with SD and ArtBank. It can be observed that our method achieves performance metrics comparable to Stable Diffusion and outperforms ArtBank. In

our quantitative experiments for different settings, SD surpasses all existing methods. Combined with visual comparison in Figure 5, SD exhibits a more pronounced stylization, which contributes to its superior performance on quantitative metrics. However, as illustrated in Figure 16, SD demonstrates limitations in disentangled representations. The generated images sometimes include elements not present in the content image prompt but instead derived from common features of the original artwork.

| Metrics | SD Rombach et al. (2022) | Artbank Zhang et al. (2024) | Ours |
|---------|--------------------------|-----------------------------|------|
| SS | 0.297 | 0.287 | 0.293 |
| IQ | 5.823 | 5.797 | 5.811 |
| TA | 0.321 | 0.291 | 0.317 |

Table 4: Quantitative comparison with stylized text-to-image generation and collection-based stylization methods.

### A.3.3 ABLATION STUDY FOR COMPONENTS IN DISENTANGLE LOSS(EQ(1))

In Eq.(1), Image-Text Contrastive Loss (ITC) primarily handles the tasks of alignment and disentanglement in the network, while the Image-Text Matching Loss (ITM) and Image-grounded Text Generation Loss (ITG) provide auxiliary support for modality alignment based on ITC. To evaluate the contribution of each loss function in training the multimodal alignment and disentanglement network, we conducted an ablation study. The results are shown in Figure 10 and Table 5, where four different configurations of the model are compared:

1) Training **with only Image-Text Contrastive Loss (ITC)**. ITC is the core loss function in the first stage, enables the model to disentangle style and content more effectively by ensuring that features corresponding to style and content are aligned with their respective textual descriptions. When the CSDN is trained exclusively with ITC, the overall content structure is preserved. When the model is trained solely with ITC, the content alignment score achieves the highest value among all configurations. This indicates that $\mathcal{L}_{itc}$ as a contrastive loss, is highly effective in aligning the content features with their corresponding textual descriptions.

However, the generated results lack intricate brushstroke details, and fail to capture the characteristic yellowish tone of traditional oil paintings, as illustrated in Figure 10(a). The lack of fine-grained alignment with styles is corroborated by the decreased SS metrics in Table 5.

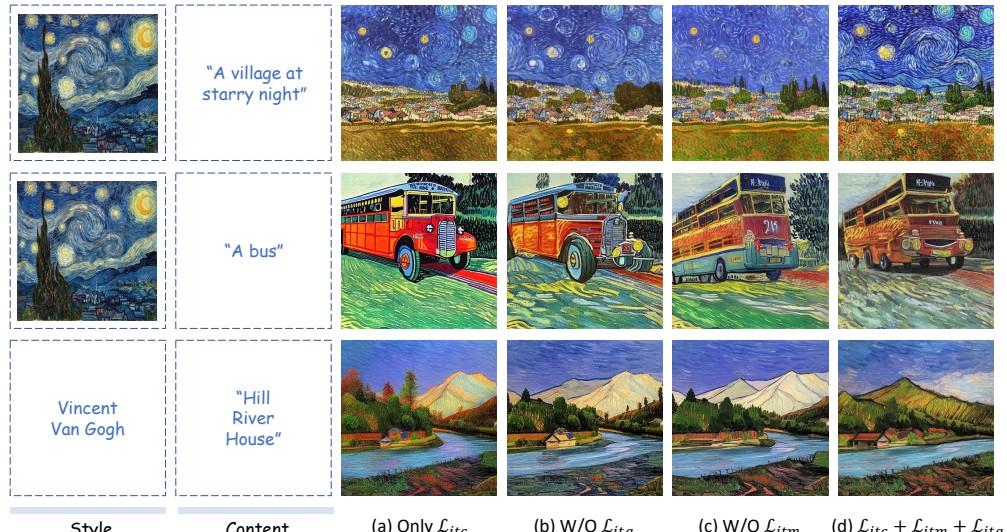

Figure 10: Visual illustration for ablation study on disentanglement loss function components.

2) Training **without Image-grounded Text Generation Loss (ITG)**: ITG trains the model to generate coherent style and content descriptions for a given image by predicting the next word or sentence

Table 5: Quantitative results for ablation study.

| Method | SS ↑ | TA ↑ |
|---|---|---|
| $\mathcal{L}_{itc}$ | 0.273 | **0.315** |
| $\mathcal{L}_{itc} + \mathcal{L}_{itm}$ (w/o $\mathcal{L}_{itg}$) | 0.279 | 0.311 |
| $\mathcal{L}_{itc} + \mathcal{L}_{itg}$ (w/o $\mathcal{L}_{itm}$) | 0.278 | 0.313 |
| $\mathcal{L}_{itc} + \mathcal{L}_{itm} + \mathcal{L}_{itg}$ (full-loss) | **0.293** | 0.308 |

based on the image and context. ITG ensures that the style and content information extracted from an image is not only disentangled but also interpretable and coherent, which enhances the overall textual-visual understanding of the model. As shown in Figure 10 (b), the absence of ITG affects the coherence of the style and content descriptions extracted from the image. For example, in the "A bus" prompt, the generated bus elements are inconsistent. Also, stylistic elements such as brushstrokes are less pronounced, leading to the drop of SS and TA in quantitative results.

3) Training **without Image-Text Matching Loss (ITM)**: ITM operates as a binary classification task, predicting whether an image-text pair is a positive or negative match. This enables the model to focus on fine-grained correspondence between images and text, such as specific objects or elements mentioned in the prompts. When ITM is removed, the generated images show a loss of detail in both style and content alignment. For instance, in Figure 10 (c), the "house" element explicitly mentioned in the content prompt is missing in the generated image. This indicates that, without ITM, the model's capacity to maintain fine-grained alignment is compromised, resulting in less coherent and contextually relevant outputs.

4) **Full-loss setting**: When all three loss functions are used together, the SS metrics achieve the highest scores. From Figure 10 (d), the generated images align closely with both style and content prompts, showcasing strong disentanglement and alignment. The style is faithfully preserved, while the content, such as the "house" or "bus," is accurately represented in the generated outputs. This demonstrates the complementary nature of the three loss functions in ensuring both disentanglement and multimodal coherence. While ITC prioritizes content alignment, the inclusion of ITM and ITG shifts the focus toward achieving a better balance, where both content and style are accurately disentangled and aligned with their textual descriptions.

### A.3.4 STYLE RESEMBLANCE COMPARISON WITH DEA-DIFF AND INSTANTSTYLE

In Figure 11, we provide a detailed comparison with InstantStyle and DEA-Diff, including the artist's original painting in the last column to offer a more intuitive demonstration of how our results more closely follow the artist's overall style. From the experimental results, InstantStyle focuses on replicating the colors and content of the reference image. DEA-Diff, on the other hand, faithfully reflects the content prompt but produces brushstrokes that deviate significantly from the reference image's style, resembling comic illustrations rather than oil paintings. In contrast, our method emphasizes imitating the artist's overall style, delivering more cohesive and stylistically accurate results. This is because DEA-Diff and InstantStyle define style differently from our approach. They focus on imitating the color and texture, whereas our method aims to replicate the artist's overall painting habits.

Under a unified style, our approach reflects different content in varied yet stylistically consistent ways. From Figure 11, regardless of the input content, DEA-Diff and InstantStyle consistently produce the same color palette and textures, which do not align with the artist's actual painting habits. On the contrary, our generated results incorporate similar colors and distinctive swirl patterns from *Starry Night* only when the input content prompt explicitly includes "starry night." Additionally, when the content prompt specifies a "city skyline," our results accurately depict urban scenes, while InstantStyle continues to replicate the rural content from the reference image. Meanwhile, DEA-Diff produces results that resemble anime rather than an oil painting. These observations demonstrate that our method effectively disentangles style and content, producing outputs that align more closely with the artist's painting habit and overall style.

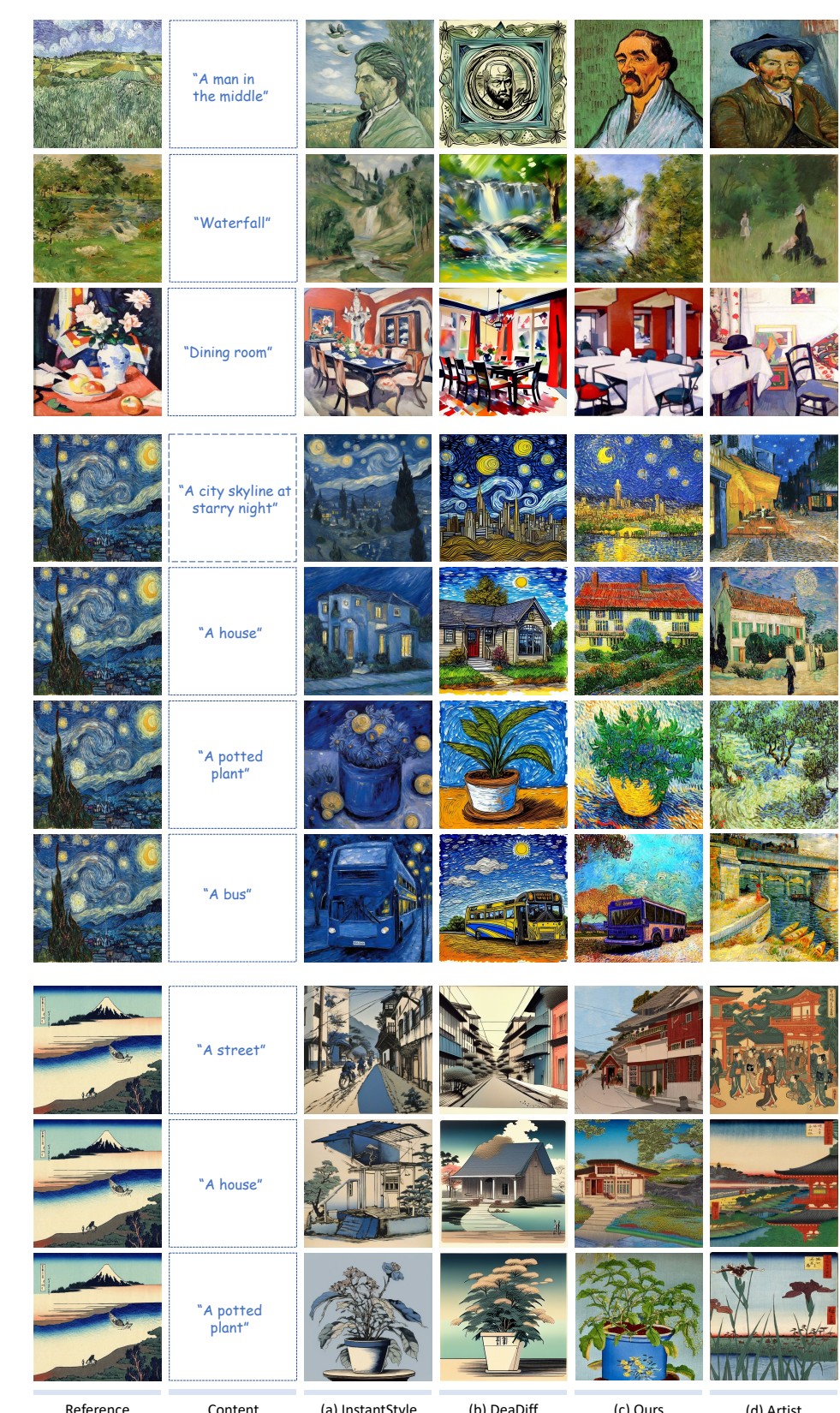

Figure 11: Visual Comparison with DEADiff and InstantStyle.

### A.3.5 IMPACT OF CONTENT AND STYLE DETAIL LEVELS ON DISENTANGLEMENT PERFORMANCE

In Figure 12, we present the impact of varying levels of detail in content and style descriptions on disentanglement performance. The descriptions for content and style can be provided either through text or images, with information complexity ranging from the simplest to the most intricate—where images provide the highest level of detail.

From subfigure (a), we observe that as the textual descriptions of content progress from simple to complex, the generated images faithfully reflect the content information described in the text. When the content is described through an artistic image, our method perfectly disentangles the content information from images, accurately capturing elements such as the flowers, vase, table, and teacup. Meanwhile, the generated image consistently mimics Monet's brushstrokes and artistic style, regardless of the level of detail in the content description.

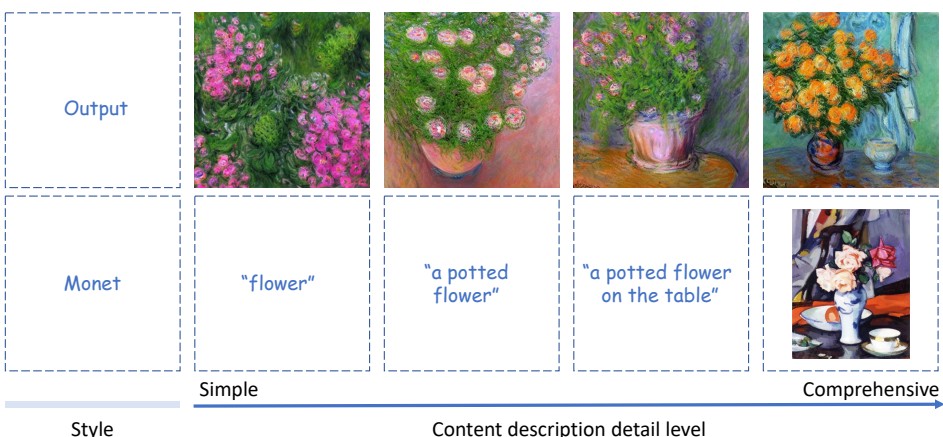

(a) Impact of content prompt detail levels on disentanglement performance

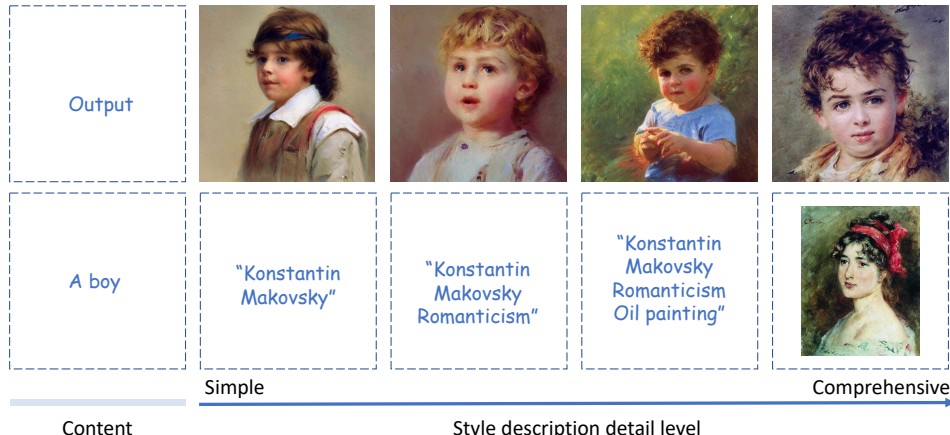

(b) Impact of style prompt detail levels on disentanglement performance

Figure 12: Impact of content and style description detail levels on disentanglement performance

From subfigure (b), we observe that as the textual descriptions of style range from a single word to the artist's actual paintings, the generated content remains entirely unaffected. However, the level of detail in the generated style increases significantly with more detailed style descriptions.

These results demonstrate the effectiveness of our network design for explicit disentanglement. The robustness of disentanglement is preserved even as the level of detail in content and style descriptions varies, further validating the reliability of our approach.

### A.3.6 COMPARISON WITH STYLEDROP

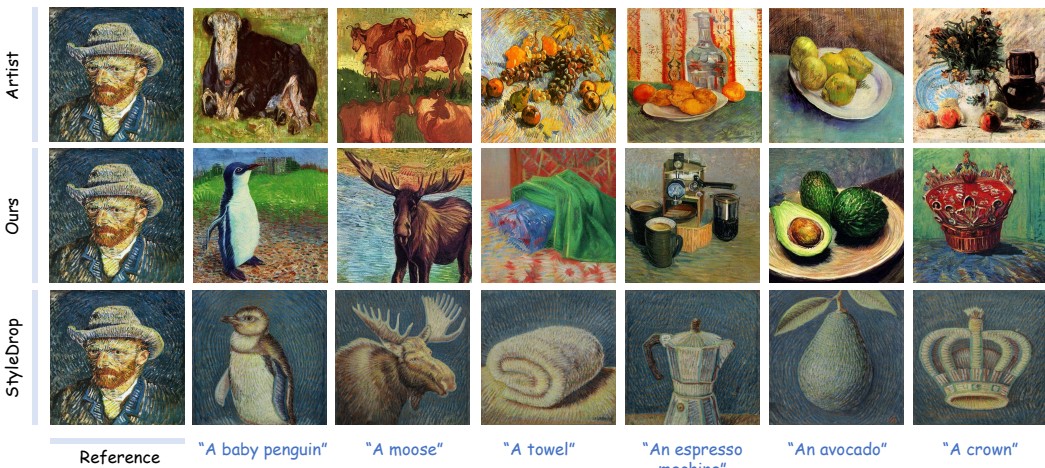

Figure 13: Visual comparison with StyleDrop.

In Figure 13, we present comparisons with StyleDrop Sohn et al. (2023). For fair comparison, we directly used the original images from their paper. The experimental results show that StyleDrop and our method have different focuses. StyleDrop fine-tunes a single reference image and content prompt, with the generated results primarily transferring the colors and textures of the reference image. In contrast, our method leverages a disentanglement design to extract style information from the reference image, capturing the artist's brushstroke techniques and overall style. This allows our approach to generate diverse outputs that align with the artist's overall style when combined with different content prompts. Additionally, we include original works by the reference artist (Van Gogh) on similar themes to the content prompts, providing a clearer demonstration that our results better follow the artist's brushstrokes and color patterns.

### A.3.7 COMPARISON WITH DREAMSTYLER

In Figure 14, we compare our method with DreamStyler Ahn et al. (2024). From the results, we observe major content leakage from the reference images in the first and second rows, where the generated results almost replicate the structure of the reference images but fail to adequately reflect the information provided by the content prompts. In the fourth row, the generated result resembles a photograph, failing to follow the style of the reference image. This limitation arises because DreamStyler focuses on enhancing content information in the generated results through text inversion during fine-tuning. As a result, when there is a significant mismatch between the content prompt and the reference style image, the disentanglement performance lacks robustness. In contrast, our method focuses on the explicit disentanglement of style and content information, achieved during the first stage of training. This approach ensures that the generated results not only faithfully adhere to the input content prompt but also closely align with the artist's overall style.

### A.3.8 DISENTANGLE PERFORMANCE COMPARISON WITH DEA-DIFF

Additionally, Fig. 15 presents a visual comparison between our method and DeaDiff. We utilize images as content prompts and various style attributes as style prompts. Both our method and DeaDiff accurately capture the theme of the content image. However, DeaDiff exhibits several failure cases in disentangle style and content information from images. For example, under the "pencil" style, the color of the mountains in the content image is incorrectly treated as style information and transferred to the generated result. With "Samuel Peploe" style, DeaDiff struggles to produce a stylized output, instead following the photographic style from the content image. These issues arise because Dea-Diff lacks the explicit disentanglement design proposed in our method, resulting in suboptimal disentanglement capability. Consequently, it often reproduces the style of the content image or the content of the style image in the generated results. In contrast, our method achieved

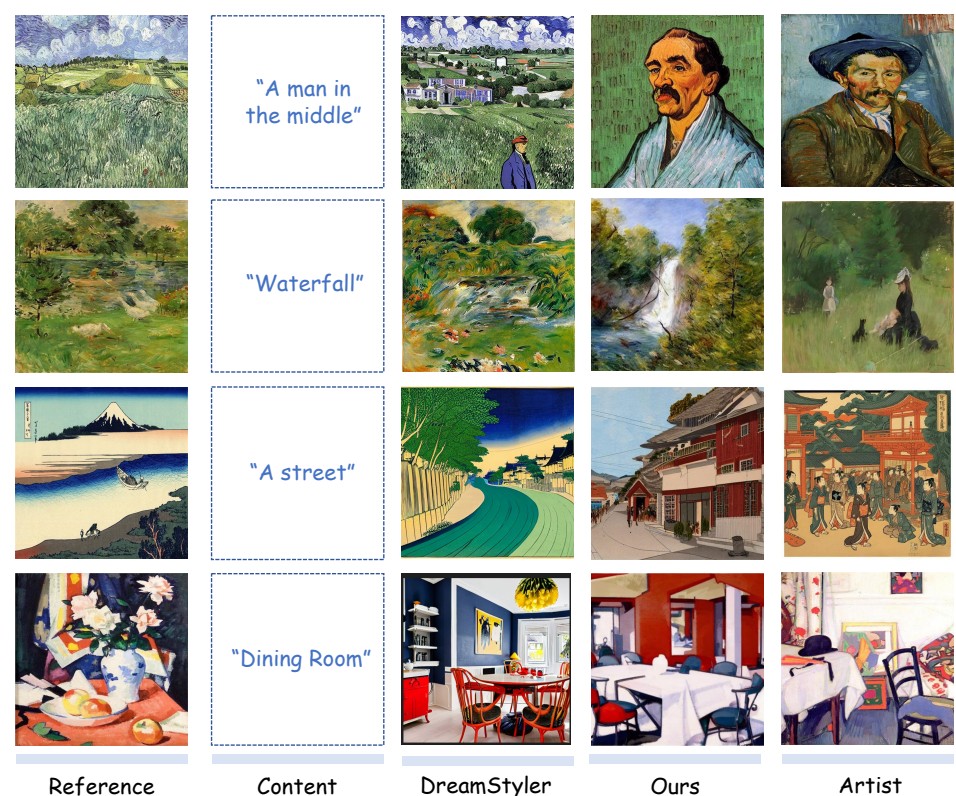

Figure 14: Visual Comparison with DreamStyler.

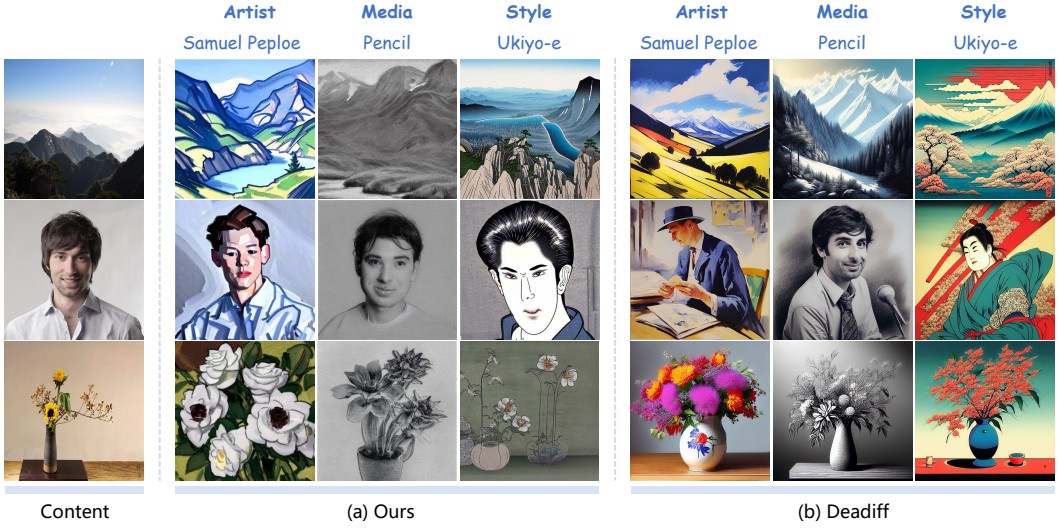

Figure 15: Content and style disentanglement comparison with DeaDiff.

robust disentanglement across scenarios where content and style inputs are provided as either images or text, while faithfully follow the artist's style.

### A.3.9 IMAGE STYLE TRANSFER

Additionally, although our model employs a non-reconstructive learning method and does not impose constraints for structural preservation, we adopt ControlNet (SoftEdge) to achieve image-based

stylization with spatial control and compared our results with existing image style transfer methods: CycleGAN (Zhu et al., 2020), F-LSeSim (Zheng et al., 2021),

Fig. 16 illustrates visual comparison with structure-preserved image style transfer. We can see that with spatial control, our method can achieve structure-preserved image style transfer. Although our model employs a non-reconstructive learning method, the generated results successfully transfer styles from reference image while maintaining the structure of the content image, avoiding the introduction of elements from the original collection that do not exist in the content image, such as the results in the fourth row from SD, InST, and Artbank.

Table 6 presents the style similarity, image quality and the overall subjective preference of our method compared with the state-of-the-art methods. We can see that our method achieves the comparable style similarity and content alignment performance. This is understandable since our method is not specifically designed for structure preservation, which would inevitably affect the visual effect of the generated results. Still, users demonstrate a significantly greater preference for our method over other ones. More detailed results and explanations could be found in appendix Sec.A.3.

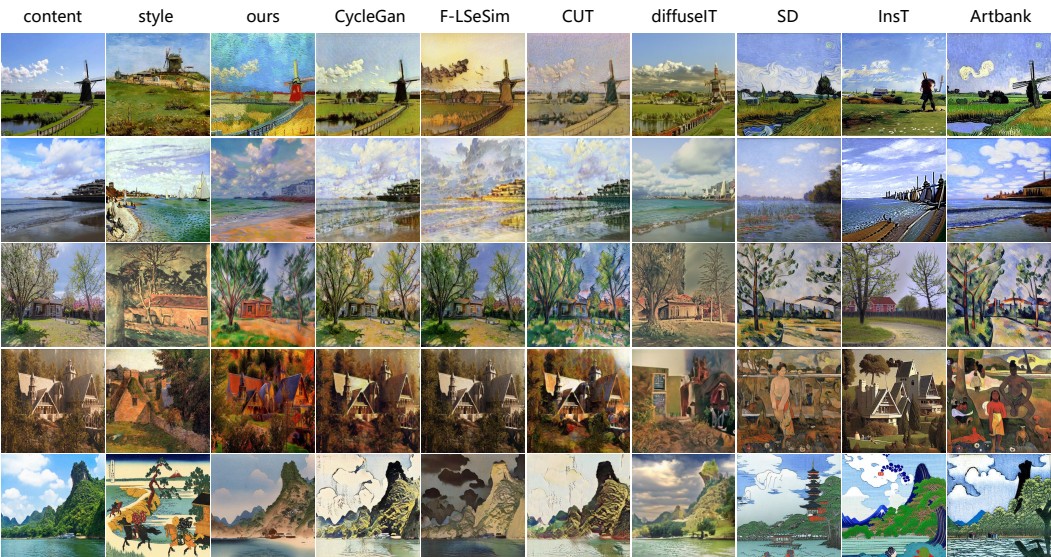

Figure 16: Qualitative comparison with image style transfer methods.

Table 6: Quantitative comparison with image style transfer methods.

| Metrics | CycleGan | CUT | F-LSeSim | DiffuseIT | SD | InsT | ArtBank | Ours |
|---------|----------|-----|----------|-----------|-----|------|---------|------|
| SS ↑ | 0.263 | 0.266 | 0.237 | 0.218 | 0.293 | **0.296** | 0.288 | 0.283 |
| IQ ↑ | 5.407 | 4.902 | 5.263 | 5.081 | 5.626 | **5.630** | 5.470 | 5.506 |
| SP ↑ | 2.778 | 3.000 | 2.556 | 2.556 | 2.833 | 2.333 | 2.761 | **3.857** |

### A.3.10 USER STUDY

Aesthetic perception to human observers is often the ultimate goal for style transfer tasks. We conduct user study to evaluate the proposed algorithm against the state-of-the-art methods. We use the most commonly mentioned 7 styles: Vincent van Gogh, Claude Monet , Paul Cezanne , Samuel Peploe , Berthe Morisot ,Paul Gauguin and Ukiyo-e. For each comparison method, we synthesized 30 images, and obtained 390 images in total. For each participants, we randomly select 35 content-style-result triplets, and display them in random order. We ask participants to rate the images to gauge their preference for these images. The scoring scale ranges from 1 to 5, where 1 represents the least liked and 5 represents the most liked. A total of 40 participants took part in our survey, rating 35 images based on their preferences.

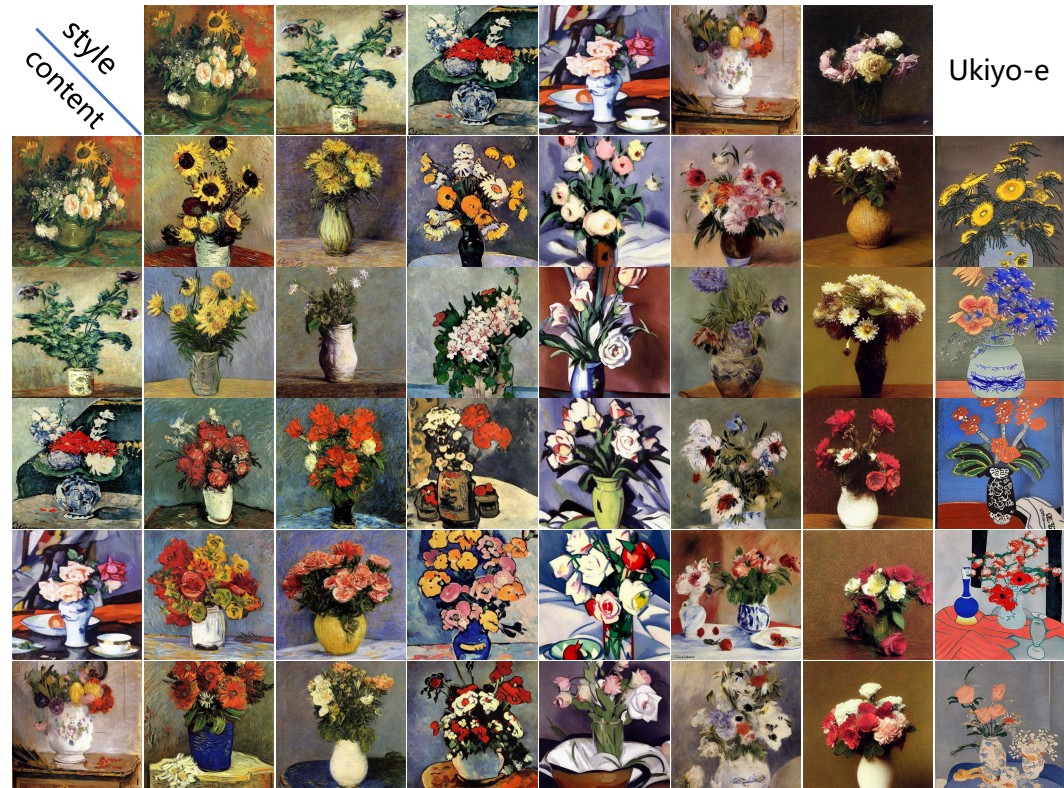

Figure 17: Qualitative results for content and style decoupling.

### A.3.11 CONTENT STYLE DISENTANGLEMENT

To verify the decoupling effect, we conducted an experiment where we use still life paintings as content and style respectively. As shown in Fig. 17, when we observe horizontally, we can see that the generated images successfully depict the themes of the content images: the first two rows features yellow flowers, the third-fifth present more red flowers. From vertical view, the generated images also effectively capture and display the style of the reference images, especially the Ukiyo-e style.

