# OpenReview forum: "Content-style disentangled representation for controllable artistic image stylization and generation"
_ICLR.cc/2025/Conference — Submitted to ICLR 2025_

### Official Review · Reviewer_gsAG · 2024-11-01

**Soundness:** 3
**Presentation:** 3
**Contribution:** 2
**Rating:** 6
**Confidence:** 4

**Summary:**

This paper addresses the challenge of artistic image stylization by decoupling content and style representations. The authors propose a new approach using a multi-modal dataset, WikiStyle+, and a disentangled content-style diffusion model guided by Q-Formers and multi-step cross-attention layers.

**Strengths:**

1. **WikiStyle+**: The introduction of the WikiStyle+ dataset is a meaningful contribution, particularly as style-content disentanglement is a growing area of interest in this research community. By providing a dataset with artwork images and descriptive text, the authors provide a resource that could aid further research into artistic stylization.

2. **Performance**: The qualitative results appear more nuanced compared to previous methods (e.g. as shown in Figure 1). For instance, in the stylization of Van Gogh’s Starry Night, the proposed method captures broader brushstroke techniques of the artist rather than simply replicating signature swirling patterns in that specific art piece. This suggests that the model emphasizes general stylistic elements of artist and it better suited for general style adaptation.

**Weaknesses:**

1. **Dataset**: The style descriptions in WikiStyle+ appear to rely on existing tags from WikiArt. This raises concerns about their expressiveness, as a simplistic style caption may not sufficiently capture the nuances of lesser-known artistic styles. Notable styles such as Van Gogh’s may embed effectively even with a very general description since the pre-trained model already have seen this artwork in their training. On the other hand, other (non-famous) styles might benefit from more detailed descriptors, such as specific brushstroke techniques or color tones, to improve style expressiveness.

2. **Novelty**: Although the authors have made commendable efforts in curating WikiStyle+, both the dataset and model contributions feel incremental. While disentangling content and style is valuable, a deeper exploration of style elements (e.g., brushstroke and color) could enhance the study’s impact (as I said in above). Similarly, the reliance on Q-Formers for modeling feels more like a marginal improvement than a breakthrough innovation.
3. **Related Works**: Recent studies e.g. StyleDrop and DreamStyler address content-style disentanglement by incorporating context prompts alongside style replication and they achieve the reduction of content leakage. Although these approaches differ in specifics, they share similarities in the goal of style-context disentanglement. A direct comparison or inclusion of these models in the discussion would clarify the unique advantages of this paper’s method. For example, it would be informative to analyze how the model performs when relying solely on style-content descriptions or to provide ablation studies to pinpoint effective components.
4. **Analysis**: It would be valuable to examine how the level of detail in content and style descriptions affects disentanglement performance. For instance, analyzing whether a minimal description results in poorer disentanglement could yield insights into the robustness of the method.

**Questions:**

Please see the weakness. I am curious on how the style replication performance varies when the details of style-content prompt differ.

---

> ### Author Response · Authors · 2024-11-29
> **Concerns about WikiStyle+**
>
> We appreciate the reviewer’s insightful comments regarding the expressiveness of style descriptions in WikiStyle+, particularly for lesser-known artistic styles. Addressing your concerns, we would like to clarify a few points:
>
> **(1) Style descriptions in WikiStyle+**
>
> The style descriptions in WikiStyle+ are indeed based on the existing tags from WikiArt, which were chosen for their scientific and objective definitions, ensuring both consistency and usability across the dataset. While these tags provide a general foundation for style representation, we acknowledge that they may lack nuanced details for lesser-known styles. we see this as a potential avenue for future exploration.
>
> While the textual descriptions of styles are relatively simple, this **does not negatively impact the experimental results**. Through the first-stage multimodal alignment pre-training, the model is able to learn features in the latent space that effectively capture the differences between artistic styles, even if the textual descriptions do not explicitly mention specific brushstrokes, compositions, or other details. Additionally, WikiStyle+ covers a **wide range of styles and artists**, enable the model to form **rich representations of artistic styles in the visual space**, compensating for the simplicity of the textual descriptions. Consequently, as demonstrated in the experimental results in Appendix 3.5, even when the textual descriptions range from detailed (e.g., an entire reference image) to simplest (e.g., a single word), the **fidelity of the generated styles does not show significant degradation**, further validating the robustness of our approach.
>
> **(2) Performance for lesser-known styles with general style descriptions**
>
> For lesser-known styles, we agree that more detailed descriptors (e.g., brushstroke techniques, color palettes) could improve expressiveness and style fidelity. Still, in our experiments, we observed that even with relatively general descriptions, our model could generate visually alike results for lesser prominent styles. Examples include **Kanstantin** (Figure7, column 4, Figure 12, (b)), **Samuel Peploe** (Figure 4, row 4; Figure 5, column 4; Figure 6, row 4), and **Berthe Morisot** (Figure 4, row 2; Figure 5, column 5; Figure 6, row 2). These results demonstrate that our model not only excels with iconic styles but also effectively learns and generates for underrepresented styles, highlighting the robustness of our approach. Moreover, as shown in the experiments in **A3.5, Figure 12 (b)**, a comparison between the fourth and fifth columns reveals that using a **general description** yields results comparable to directly **using the style image** as the description.
>
> Still, the experiments in A3.5 indeed demonstrate a clear trend: **more detailed descriptions can enhance the style expressiveness of the results**. To address this and further explore the potential of detailed style description, we plan to extend the style descriptions in WikiStyle+ as our future work by incorporating more specific attributes , such as brushstroke details, compositional techniques, and color schemes.

---

> ### Author Response · Authors · 2024-11-29
> **Novelty**
>
> While content-style disentanglement has been explored in prior works, we significantly advanced this field by achieving explicit disentanglement of content and style at the feature level, enabling us to capture the overall style of the artist. This progress is attributed to our **innovative contributions in both dataset construction and model design**. Specifically:
>
> **(1) Contributions of WikiStyle+**:
>
> WikiStyle+ is more than a curated dataset—it is designed to address the challenge of disentangling content and style, a foundational problem in artistic stylization. Compared to existing datasets, WikiStyle+ introduces:
>
> - A significantly larger scale, encompassing over 146k images and 209 styles, which broadens its applicability for various downstream tasks.
> - The construction of **artistic image–style description–content description triplets**, making it the **first dataset of its kind**. This structure not only supports disentanglement but also enables multimodal tasks such as style transfer, retrieval, and generation.
>
> While incremental contributions to datasets are often perceived as less innovative, we believe WikiStyle+ **fills a critical gap by emphasizing disentanglement**, and our experimental results highlight its robustness even for lesser-known styles with general descriptions.
>
> **(2) Contributions of CSDN**:
>
> Although previous approaches to content-style disentanglement have utilized Q-Formers, their disentanglement methods were not explicit. Instead, they typically relied on synthesized image pairs (DEA-Diff) or assumed linear additivity between content and style features in the latent space (InstantStyle), which often resulted in noticeable content leakage in the generated outputs. To the best of our knowledge, no framework has yet been proposed to achieve **explicit content-style disentanglement at the feature level**. Our work is the first to introduce a framework specifically designed for explicit disentanglement by multimodal alignment , highlighting its robustness in comprehensive experiments.
>
> - **Furthermore, our approach goes beyond simply integrating Q-Formers:**
>
>   * We carefully integrate Q-Formers into a novel pipeline to achieve explicit disentanglement at the feature level. This is non-trivial, as it requires aligning features across modalities while preserving both content and style fidelity.
>   * The disentanglement process leverages both WikiStyle+ and Q-Formers to achieve significant improvements in robustness and flexibility. The robustness of our disentanglement approach is validated by the consistent generation quality across varying levels of detail in input descriptions, which discussed in Appendix 3.5, Figure 12. Moreover, our method is capable of disentangling content not only from photographs but also from artistic images, effectively separating content and style, as demonstrated in Figures 7 and 17.
>
> We agree that further exploration of specific style elements, such as brushstrokes and color palettes, is indeed an exciting direction. We thank the reviewer for highlighting these areas of improvement, as they align with our long-term goals for advancing the field of artistic stylization. While our current work focuses on building a robust foundation for content-style disentanglement, we see it as a stepping stone toward deeper and more granular exploration of artistic styles.

---

> ### Author Response · Authors · 2024-11-29
> **Comparison with Related Works**
>
> In Appendix Figures 14 and 15, we present comparative experiments with DreamStyler and StyleDrop.
>
> - Comparison with StyleDrop
>
> For the comparison with StyleDrop, we used the original results provided in its paper. The experimental results demonstrate that StyleDrop and our method have distinct focuses. StyleDrop does a good job in **transferring the color and texture of the reference image** to the generated results by fine-tuning with a single reference image and a content prompt. In contrast, our method employs a disentangled design to extract the style characteristics from the reference image, capturing the **artist's overall style**. This allows our method to generate **diverse results that remain consistent with the artist's style**. To more intuitively highlight our method's ability, we included original works by the reference artist (Van Gogh) with themes similar to the content prompts. The results further validate that our method achieves higher consistency with the artist's distinctive style compared to StyleDrop. Under a unified style, our approach reflects different content in varied yet stylistically consistent ways.
>
> - Comparison with DreamStyler
>
> For comparison with DreamStyler, we observed major content leakage from the reference images, where **the generated results almost replicate the structure of the reference images but fail to adequately reflect the content prompts**. In the fourth row, the generated result resembles a photograph, failing to follow the style of the reference image. This limitation arises because DreamStyler focuses on enhancing content information in the generated results through text inversion during fine-tuning. As a result, when there is a significant mismatch between the content prompt and the reference style image, the disentanglement performance drops. In contrast, our method focuses on the explicit disentanglement of style and content features. This approach ensures that the generated results **not only faithfully adhere to the input content prompt but also closely align with the artist's overall style**.
>
> In conclusion, while both DreamStyler and StyleDrop produce intriguing results, their core focus and methodologies differ significantly from ours. DreamStyler and StyleDrop primarily emphasize replicating the color and texture of the reference image by leveraging text inversion techniques during the fine-tuning stage. In contrast, our research focuses on achieving a more effective disentanglement of content and style from reference images, aiming to emulate the artist’s painting style—characterized by **diverse yet consistent artistic expressions across different subjects**.

---

> ### Author Response · Authors · 2024-11-29
> **How the level of detail in content and style descriptions affects disentanglement performance**
>
> In Appendix Section 3.5, Figure 12, we present an analysis of the impact of varying levels of detail in content and style descriptions on disentanglement performance. The results demonstrate the robustness of our method across a wide range of detail levels.
>
> - **Changes in Content Descriptions**:
>
>   As illustrated in Figure 12(a), whether the content description is as simple as a single word or as detailed as an image, our method achieves good disentanglement of the content information. The generated images **accurately capture the described content, whether from minimal or detailed descriptions, while maintaining consistent stylistic alignment with the artist**. This demonstrates the model's ability to effectively capture content semantics across varying levels of descriptive detail.
>
> - **Changes in Style Descriptions**:
>
>   As shown in Figure 12(b), as the style descriptions progress from single words to detailed depictions like an artist’s paintings, the **brushstrokes and styles in the generated images become increasingly nuanced and refined. However, the themes of the generated images remain unaffected**, demonstrating the model’s strong disentanglement capability.
>
> These results highlight the robustness of our explicit disentanglement network design, as it consistently maintains stable disentanglement performance across varying levels of detail in content and style descriptions. Even with minimal descriptions, the model reliably achieves disentanglement. Richer style descriptions enhance the stylistic expressiveness of the generated results, while more detailed content descriptions result in correspondingly richer content in the output. However, **these descriptions only affect their respective aspects**, and variations in the amount of descriptive information do not impact the disentanglement performance.

---

> > ### Comment · Reviewer_gsAG · 2024-12-02
> >
> > Thanks for the reply. For me, the author rebuttal addresses most of my concerns and questions, and especially, Figure 12 and A3.5 seem to show interesting results. Hence, I decided to increase my rating.
> >
> > However, regarding the comparison with DreamStyler, in my experience with that model, I think the content leakage in DreamStyler might be related to not providing a sufficiently long context prompt. While the fixed settings used in this paper for comparison make this unavoidable, I suspect that the prompts used may differ slightly from those suggested in the DreamStyler paper.

---

> > > ### Author Response · Authors · 2024-12-04
> > > **Author's Reply**
> > >
> > > Thank you for taking the time to review and consider our responses!
> > >
> > > In our comparison with DreamStyler, we used the same template as described in the paper. DreamStyler utilizes **_"a painting + contextual prompt + in the {} style"_** for finetuning. The contextual prompt serves as a description of the content in the style image and {} represents the reference image. Similarly, in our experiments, **we constructed prompts following the same structure**. For instance, in the 1st row of Figure 14, the contextual prompt is *"wheatfield landscape”* so **the prompt we used for finetune DreamStyler is “*A painting of a wheat field landscape in the {} style* “**.
> > >
> > > In our experiment, we utilized a multimodal large language model (LLM) to generate content description for the reference image, following our dataset construction methodology. However, as noted in DreamStyler, automatic prompt construction alone does not completely resolve content leakage. While integrating human feedback into the prompt design process is considered the most effective approach, manual generation of prompts is challenging due to the lack of a consistent standard. To ensure uniformity, we standardized the process by relying on LLM-generated prompts.
> > >
> > > This **LLM-generated contextual prompt may contribute to the content leakage observed in DreamStyler**, as contextual prompts refined with human feedback would likely be more detailed and yield better results. Some of the contextual descriptions we used are sufficiently detailed, such as the 3rd one, which alleviated the content leakage from DreamStyler.
> > >
> > > Here are the four contextual prompts we used for comparison with DreamStyler, which will be supplemented in the corresponding section of the Appendix later.
> > >
> > > 1) "A painting of a wheat field landscape in the style of {}."
> > >
> > > 2) "A painting of a serene park scene with people rowing boats on a lake and geese on the grass in the style of {}."
> > >
> > > 3) "A painting of Mount Fuji viewed from the sea with gentle waves, a small boat carrying goods, and a people and a horse walking along the shore in the style of {}."
> > >
> > > 4) "A painting of a still life arrangement with a vase of flowers, a bowl of fruit, and some fabric in the style of {}."

---

### Official Review · Reviewer_fLvE · 2024-11-01

**Soundness:** 3
**Presentation:** 3
**Contribution:** 4
**Rating:** 5
**Confidence:** 3

**Summary:**

This paper presents an innovative approach to controllable artistic image stylization and generation by addressing the challenges of content and style disentanglement. By constructing the WikiStyle+ dataset, which includes artworks with corresponding textual descriptions, the authors enable a more comprehensive disentanglement of content and style. Their proposed model utilizes Q-Formers and learnable multi-step cross-attention layers within a pre-trained diffusion model, allowing for inputs from different modalities. The experimental results demonstrates that the method achieves thorough disentanglement and harmonious integration of content and style. This work represents a significant advancement in the field.

**Strengths:**

- This paper creatively proposes the use of a Q-Former to disentangle content and style features, achieving more fine-grained control in generation and yielding excellent results.
- The paper conducts thorough comparisons, surpassing baseline methods on multiple metrics, and provides ample visual analyses to support its main conclusions.
- The writing is fluent, the expressions are clear, and the logic is easy to follow.
- The primary research problem of this paper—disentangling content and style features—offers valuable insights for the development of related fields.

**Weaknesses:**

Some of the description of the methods is unclear:

- MCL is mentioned in Line 251 but is not further elaborated upon.
- In Eq 3, the specific workings of the binary classification network are not described, including its specific inputs. The text generation model used in Eq 4 is not introduced in the text or figures.


The mathematical expressions and symbols are ambiguous:

- In Eq(2), the subscripts c and s on the vectors I and T previously indicated different feature types, but in the equation, i is used to denote the sample index.
- The symbol T has multiple meanings: in Figure 4, it represents the total number of forward pass steps; in Eq 2, it denotes text features; and in Eq 4, it indicates text length. Similarly, t has multiple meanings, referring to both timestep and text sequence index.
- Image features are denoted as z in Fig 4, but as F_I in the text.

There are noticeable typos in the keywords:

- Line 251: "Mlti-step Cross-attention Layers" should be "Multi-step Cross-attention Layers."
- Line 337: "learning rate of 51e-5" should be corrected.
- Fig 4: "Detangle Loss" should be "Disentangle Loss."

**Questions:**

- What are the differences between CSDN and MCL? Does CSDN refer to the entire generation framework or specifically to the cross-attention part?
- Could you provide a more detailed analysis of the roles of each component in the disentanglement loss function, such as through mor ablation experiments?

---

> ### Author Response · Authors · 2024-11-29
> **Description of the Methods**
>
> - **MCL is mentioned in Line 251 but is not further elaborated upon.**
>
> We revised the writing in Section 4.2 to include a more detailed explanation of MCL:  “We use multi-step learnable cross-attention layers (MCL) to inject the style embeddings into the denoising process of the SD model. At each timestep of the diffusion process, the style features are introduced as conditions through the cross-attention layers in MCL to guide the generation process. These cross-attention layers embed the style features into the current diffusion features using the attention mechanism.”
>
>  For a more comprehensive context, please refer to Section 4.2 of the revised PDF.
>
> - **In Eq 3, the specific workings of the binary classification network are not described, including its specific inputs.**
>
> In Eq. (3), the ITM loss is formulated as binary classification task, **predicting whether an image-text pair is a match (positive) or not (negative)**. This encourages the model to focus on fine-grained correspondence between images and text. The binary classifier is updated alongside the main parameters of the Q-Former during the ITM task.
>
> The ITM loss computes the cosine similarity between the image embedding $I$ and text embedding $T$, and then uses a linear layer to map the cosine similarity to a matching probability. For content correspondence matching, **the inputs are ($I_c$ , $T_c$)**; for style correspondence matching, **the inputs are ($I_s$ , $T_s$)**. The ITM loss employs a binary classification loss to optimize both the Q-Former and the classifier. We included a more detailed introduction for ITM task in Sec.4.1 of the revised PDF (L260-269).
>
> - **The text generation model used in Eq 4 is not introduced in the text or figures.**
>
> We use a lightweight text decoder as text generation model in ITG loss. Its primary structure includes a transformation module that applies a dense projection, an activation function, and layer normalization to refine hidden states, and a decoder layer that maps the processed hidden states to vocabulary logits using a linear layer. We included a more detailed introduction for ITG task in Sec.4.1 of the revised PDF (L270-279).

---

> ### Author Response · Authors · 2024-11-29
> **Mathematical expressions and symbols & Typos**
>
> - **Mathematical expressions and symbols**
>
>   In the revised PDF, we made the following modifications for mathematical expressions and symbols:
>
>   * Throughout the text, we use the uppercase letter **$I$** to represent the **image modality**. In Equation (2), we replaced I with the lowercase letter **$n$ to denote the sample index**. Accordingly, modifications were also made to Equation (3).
>   * We standardized the notations across the text. Following common conventions in the literature, we use the lowercase letter **$t$ to denote the time step**, the uppercase letter **$T$ to represent text** (Equation 2), and **$M$ and $m$ to denote text length and text sequence index**, respectively (Equation 4). The corresponding sections of the text have been updated to incorporate these changes.
>   * $Z$ and $Z_t$ are not image features; they represent the **noise states at time step $t$** during the diffusion process. We updated the framework diagram (now Figure 3) to provide a clearer depiction of the diffusion process module. We use $Z_0$ and $Z_T$ in Figure 3 to denote noise states at the first and last time step, respectively.
>
> - **Typos**
>
>   We greatly appreciate your attention to detail. We have carefully reviewed the text and corrected all identified typos in the revised version. Additionally, we have conducted a thorough proofreading of the entire manuscript to ensure that no other typos remain.

---

> ### Author Response · Authors · 2024-11-29
> **Differences between CSDN and MCL**
>
> **CSDN is the network trained in the first stage of our method**, with the objective of aligning the style features of artistic images with style description texts and the content features with content description texts in the feature space, thereby achieving the disentanglement of style and content. The **structure of CSDN** includes a pre-image encoder and a Q-Former, where the Q-Former is the trainable module designed to extract multimodal representations related to style and content (disentanglement). The training of CSDN involves three pretraining tasks: Image-Text Contrastive Learning (ITC), Image-Text Matching (ITM), and Image-Text Generation (ITG). The Q-Former consists of main parameters, a binary classifier for ITM, and a text decoder for ITG. The main parameters are jointly updated by all three tasks, while the binary classifier and text decoder are updated specifically by the ITM and ITG losses, respectively.
>
> **MCL is the structure that injects the disentangled embeddings from CSDN as control conditions into the diffusion process** at different timesteps using cross-attention mechanisms. At each timestep of the diffusion process, the style and content embeddings are introduced as conditions through the cross-attention layers in MCL to guide the generation process.
>
> We have provided a more detailed explanation of cross-attention in MCL in section 4.2 in the revised PDF. Additionally, we have **made the distinction between CSDN and MCL more explicit in the framework figure** (Figure 3) to better highlight their differences.

---

> ### Author Response · Authors · 2024-11-29
> **Ablation Study for Each Component in Disentanglement Loss Function**
>
> In disentanglement loss function, **Image-Text Contrastive Loss (ITC)** primarily handles the tasks of alignment and disentanglement in the network, while the **Image-Text Matching Loss (ITM)** and **Image-grounded Text Generation Loss (ITG)** provide auxiliary support for modality alignment based on ITC. To evaluate the contribution of each loss function and their impact on final results, we conducted **an ablation study in Appendix 3.3**, where four different configurations of the model are compared:
>
> 1. Training **with only Image-Text Contrastive Loss (ITC)**. ITC is the core loss function in the first stage, enables the model to disentangle style and content more effectively by ensuring that features corresponding to style and content are aligned with their respective textual descriptions.
> When the CSDN is trained exclusively with ITC, the overall content structure is preserved. When the model is trained solely with ITC, the content alignment score achieves the highest value among all configurations. This indicates that $\mathcal{L}_{\mathit{itc}}$ as a contrastive loss, is highly effective in aligning the content features with their corresponding textual descriptions. However, the generated results lack intricate brushstroke details, and fail to capture the characteristic yellowish tone of traditional oil paintings, as illustrated in **Fig. 10(a)**. The lack of fine-grained alignment with styles is corroborated by the decreased SS metrics in **Table 5**.
> 2. Training **without Image-grounded Text Generation Loss (ITG)**: ITG trains the model to generate coherent style and content descriptions for a given image by predicting the next word or sentence based on the image and context. ITG ensures that the style and content information extracted from an image is not only disentangled but also interpretable and coherent, which enhances the overall textual-visual understanding of the model. As shown in **Fig.10 (b)**, the absence of ITG affects the coherence of the style and content descriptions extracted from the image. For example, in the "A bus" prompt, the generated bus elements are inconsistent. Also, stylistic elements such as brushstrokes are less pronounced, leading to the drop of SS and TA in quantitative results.
> 3. Training **without Image-Text Matching Loss (ITM)**: ITM operates as a binary classification task, predicting whether an image-text pair is a positive or negative match. This enables the model to focus on fine-grained correspondence between images and text, such as specific objects or elements mentioned in the prompts. When ITM is removed, the generated images show a loss of detail in both style and content alignment. For instance, in **Fig. 10 (c)**, the ``house'' element explicitly mentioned in the content prompt is missing in the generated image. This indicates that, without ITM, the model’s capacity to maintain fine-grained alignment is compromised, resulting in less coherent and contextually relevant outputs.
> 4. **Full-loss setting**: When all three loss functions are used together, the SS metrics achieve the highest scores. From **Fig. 10 (d)**, the generated images align closely with both style and content prompts, showcasing strong disentanglement and alignment. The style is faithfully preserved, while the content, such as the "house" or "bus," is accurately represented in the generated outputs. This demonstrates the complementary nature of the three loss functions in ensuring both disentanglement and multimodal coherence. While ITC prioritizes content alignment, the inclusion of ITM and ITG shifts the focus toward achieving a better balance, where both content and style are accurately disentangled and aligned with their textual descriptions.
>
> In conclusion:
>
> - **ITC** ensures effective alignment of visual and textual features.
> - **ITG** ensures that the style and content information extracted from an image is not only disentangled but also interpretable and coherent. Without ITG, the generated images may lose stylistic fidelity or fail to fully capture content prompts, as the model would lack a strong textual grounding during training.
> - **ITM** enhances fine-grained alignment between textual descriptions and corresponding visual elements, enabling the model to accurately reflect details in the generated images. Without ITM, the relevance of generated images to the input prompts degrades, with style and content features becoming less accurately matched to their textual descriptions.

---

> > ### Comment · Reviewer_fLvE · 2024-12-02
> >
> > Thanks for the reply, after careful consideration, I decided to maintain my rating

---

> > > ### Author Response · Authors · 2024-12-03
> > > **Author's reply**
> > >
> > > Thank you for taking the time to review and consider our responses. We would like to kindly confirm if our responses have adequately addressed your concerns. If there are any remaining issues or additional feedback, we would be happy to address them.
> > >
> > > Thank you once again for your time and effort in reviewing our manuscript and for providing your thoughtful feedback!

---

### Official Review · Reviewer_ikfg · 2024-11-02

**Soundness:** 2
**Presentation:** 3
**Contribution:** 2
**Rating:** 5
**Confidence:** 4

**Summary:**

This paper proposes a content-style representation disentangling method for controllable artistic image stylization and generation. The proposed method employs contrastive learning tasks to learn disentangled content and style representations, which then guide a diffusion model to generate stylized images.

**Strengths:**

1. The proposed method accepts inputs from different modalities as control conditions.
2. This paper provides a new dataset consists of artworks with corresponding textual descriptions for style and content.
3. Both qualitative and quantitative experiments are conducted to evaluate the performance of the proposed method.

**Weaknesses:**

1. The proposed method is not well-explained. What are the outputs and training objectives of the Content and Style Disentangled Network (CSDN)? What is its structure? The paper states, 'The image-grounded text generation loss involves training a model to generate descriptive text that corresponds to a given input image.' To whom does the 'model' refer in this context? Additionally, what is the 'two-class linear classifier' mentioned in the image-text matching loss, and where does it come from?

2. The claims regarding the quality of the proposed method in the text-to-image stylization task appear to be overstated. While its visual quality is comparable to that of other methods, it is challenging to identify instances where the proposed method demonstrates significant superiority. In fact, the stylized images produced in this paper always exhibit noticeable deviations in style (such as color) when compared to the reference style images.

3. Some state-of-the-art text-to-image stylization methods are not compared in this paper, such as StyleDrop [1] and DreamStyler [2]. \
[1] StyleDrop: Text-to-Image Generation in Any Style. NeurIPS 2023. \
[2] DreamStyler: Paint by Style Inversion with Text-to-Image Diffusion Models. AAAI 2024.

4. The style similarity metric used in this paper lacks persuasiveness. Why not employ CLIP to directly assess the similarity between the generated images and the reference images? Alternatively, Gram loss is also a widely used metric to evaluate style similarity between two images.

5. This paper only conducted quantitative experiments in the text-to-image stylization task, while no quantitative experiments were performed in the stylized text-to-image generation and collection-based stylization tasks.

6. I am curious about the inference speed of the proposed method. Is it comparable to or superior to that of previous methods?

**Questions:**

Please see **Weaknesses**.

---

> ### Author Response · Authors · 2024-11-29
> **Method Explanation**
>
> - What are the outputs and training objectives of the Content and Style Disentangled Network (CSDN)? What is its structure?
>
>   * The **output of CSDN consists of disentangled style and content embeddings**. The training objective of CSDN is to align the style features of artistic images with style descriptions and the content features of images with content descriptions in the feature space, thereby achieving disentanglement of style and content. **The structure of CSDN includes a pre-image encoder and a Q-Former**. The Q-Former, as a trainable module, is designed to explicitly disentangle features by extracting multimodal representations related to style and content, respectively.
>
>   * CSDN is trained using three pretraining tasks: Image-Text Contrastive Learning (ITC), Image-Text Matching (ITM), and Image-Text Generation (ITG). The trainable parameters of the Q-Former include the main module parameters, the binary classifier parameters for ITM, and the text decoder parameters for ITG. Among these, the main module parameters are updated jointly by all three tasks, while the binary classifier and text decoder are updated specifically by the ITM and ITG losses, respectively.
>
>   * To enhance clarity, we optimized the framework figure by **including some intermediate variables to make the structure of CSDN more explicit**.
>
> - The image-grounded text generation loss involves training a model to generate descriptive text that corresponds to a given input image. To whom does the 'model' refer in this context?
>   * Here, the ‘model’ refers to CSDN, specifically the **main parameters of the Q-Former and the text decoder** updated through the ITG task in first stage training. A lightweight text decoder is used to generation the text sequence. It consists of two main components: a transformation module that applies a dense projection, an activation function, and layer normalization to refine hidden states, and a decoder layer that maps the processed hidden states to vocabulary logits using a linear layer. We have provided a more detailed introduction of ITG in Section 4.1 of the revised PDF.
>
> - What is the 'two-class linear classifier' mentioned in the image-text matching loss, and where does it come from?
>   * In the image-text matching loss, the 'two-class linear classifier' refers to **an auxiliary binary classifier that predicts whether an image-text pair is a match (positive) or not (negative)**. This binary classifier is updated alongside the main parameters of the Q-Former during the ITM task. ITM computes the cosine similarity between image embedding and text embedding, then using a linear layer to map the cosine similarity into matching probability. ITM uses binary classification loss to optimize the Q-former and classifier. **As one of the three training tasks for CSDN, the ITM task focuses on learning fine-grained multimodal relationships between images, style text, and content text**. This enables the model to focus on fine-grained correspondence between images and text. We have provided a more detailed introduction of ITM in Section 4.1 of the revised PDF.

---

> ### Author Response · Authors · 2024-11-29
> **Stylization Performance**
>
> We understand that the perception of style similarity can vary among individuals, which may explain the reviewer’s concern that our generated results deviate from the reference style (e.g., in terms of color). While color similarity is one interpretation of style correlation, our primary goal is to emulate the artist’s overall style—specifically, the **diverse yet consistent artistic expressions when portraying different themes**. We would like to clarify that our statement of result quality is based on our aim to capturing the broader stylistic essence of an artist for stylization,  rather than transferring colors. Under a unified style for an artist or genre, our approach produces results that are diverse yet remain stylistically consistent.
>
> Achieving this objective relies on disentanglement, which is why we also place significant emphasis on **comparing performance in terms of disentanglement** when analyzing the results. While some methods succeed from the perspective of color preservation, they often **fail to generate results that faithfully align with the content prompt**. Take Figure 4 (formerly 5) for example, our method achieves the best balance between generating content consistent with the content prompt and adhering to the style of the artist represented by the reference image.
>
> We would also like to highlight the observations made by two other reviewers, fLvE and gsAG, who noted that our method **“captures the nuanced brushstroke techniques of artists rather than simply replicating signature swirling patterns”** (Figure 1) or merely transferring colors (Figures 4 and 6), and **“achieved more fine-grained control in generation, yielding excellent results”** . These comments align with our definition of style, which goes beyond simple replication of color or texture, and further validate our claims about the effectiveness of our approach.
>
> Although InstantStyle and DEA-Diff also aim to disentangle content and style, their generated results often reproduce the style of the content image or the content of the style image, indicating insufficient disentanglement. Specifically, the brushstrokes of DEA-Diff resemble those of comic art rather than oil paintings, while InstantStyle exhibits content leakage from the reference image. For example, in the first row, the generated image includes elements such as wheat fields and trees from the reference image, even though these are not mentioned in the content prompt. Similarly, IP-Adapter and T2I-Adapter maintain style consistency but fail to successfully generate content specified in the prompt.
>
> To further support our explanation, we have included additional visual experiments in Appendix Figure 11. It is important to note that our method is **capable of transferring colors and textures**. However, much like emulating an artist’s painting style, this occurs only when the content prompt is thematically aligned with the reference image. In such cases, our method produces colors and textures that closely match those of the reference image.

---

> > ### Comment · Reviewer_ikfg · 2024-12-03
> >
> > Given just a single reference image, how can the authors capture the artist's overall style? And how do the authors even define what the artist's 'overall style' is? I argue that this goal is unrealistic. A more reasonable objective would be to focus on emulating the style of the reference image itself—its colors, textures, brushstrokes, and so on.

---

> > > ### Author Response · Authors · 2024-12-04
> > > **Author's Reply**
> > >
> > > Thank you for taking the time to review and consider our responses!
> > >
> > > - **A single reference image is not the sole source of style for the generated results; it also includes the style patterns embedded in the pretraining dataset.** During the pretraining phase, the model learns and clusters various style features from a large-scale dataset (e.g., 1. Artists who paint blue skies often depict sunflowers in gold; 2. Artists who favor short, distinct brushstrokes tend to use similar techniques across their works). During inference, the model extracts style information from the single reference image and combines it with the knowledge from the pretraining model to infer the target style.
> > >
> > > - In other words, this work does not simply transfer all style elements from a single reference image indiscriminately. For different content prompts, some style elements in the single image may be reasonable, while others may not fit the context. **The proposed model learns to to make appropriate style inferences based on different content prompts, effectively filtering out unreasonable style elements**, resulting in more natural and authentic outputs. For example, when the reference image is Van Gogh's Starry Night, the generated results mimic the brushstrokes from the reference image while filtering out unreasonable colors. Otherwise, if the target content prompt is "sunflowers," would generating "blue sunflowers sparkling with stars" truly represent Van Gogh's style?
> > >
> > > We hope this response addresses the reviewer's concern.

---

> ### Author Response · Authors · 2024-11-29
> **Comparison with DreamStyler and StyleDrop**
>
> In Appendix Figures 14 and 15, we present comparative experiments with DreamStyler and StyleDrop.
>
> - Comparison with StyleDrop
>
> For the comparison with StyleDrop, we used the original results provided in its paper. The experimental results demonstrate that StyleDrop and our method have distinct focuses. StyleDrop does a good job in **transferring the color and texture of the reference image** to the generated results by fine-tuning with a single reference image and a content prompt. In contrast, our method employs a disentangled design to extract the style characteristics from the reference image, capturing the **artist's overall style**. This allows our method to generate **diverse results that remain consistent with the artist's style**. To more intuitively highlight our method's ability, we included original works by the reference artist (Van Gogh) with themes similar to the content prompts. The results further validate that our method achieves higher consistency with the artist's distinctive style compared to StyleDrop. Under a unified style, our approach reflects different content in varied yet stylistically consistent ways.
>
> - Comparison with DreamStyler
>
> For comparison with DreamStyler, we observed major content leakage from the reference images, where **the generated results almost replicate the structure of the reference images but fail to adequately reflect the content prompts**. In the fourth row, the generated result resembles a photograph, failing to follow the style of the reference image. This limitation arises because DreamStyler focuses on enhancing content information in the generated results through text inversion during fine-tuning. As a result, when there is a significant mismatch between the content prompt and the reference style image, the disentanglement performance drops. In contrast, our method focuses on the explicit disentanglement of style and content features. This approach ensures that the generated results **not only faithfully adhere to the input content prompt but also closely align with the artist's overall style**.
>
> In conclusion, while both DreamStyler and StyleDrop produce intriguing results, their core focus and methodologies differ significantly from ours. DreamStyler and StyleDrop primarily emphasize replicating the color and texture of the reference image by leveraging text inversion techniques during the fine-tuning stage. In contrast, our research focuses on achieving a more effective disentanglement of content and style from reference images, aiming to emulate the artist’s painting style—characterized by **diverse yet consistent artistic expressions across different subjects**.

---

> ### Author Response · Authors · 2024-11-29
> **Style Similarity Metric**
>
> - **Clarification on selected metric**
>
> The style similarity metric we selected is from DEA-Diff (CVPR 2024). The calculation of this metric involves first generating descriptive text for the reference image and then removing content-related elements from the description. The style similarity is then calculated by computing the CLIP similarity between the generated image and the style description text.
>
> We chose not to directly measure the CLIP similarity between the generated image and the reference image for two main reasons:
>
> 1. This metric is not calculated directly using the CLIP features of the reference image but rather based on the CLIP features of the style description derived from it. This ensures that the **similarity is measured specifically between the generated results and the style characteristics of the reference image**, rather than its overall similarity. Consequently, this metric is better suited for evaluating a model's ability to disentangle content and style in multimodal (text-to-image) tasks, aligning with the core objective of our approach, as well as that of DEA-Diff, ArtBank.
> 2. Since CLIP is pre-trained primarily for image recognition tasks, directly using the CLIP similarity between the reference image and the generated image would predominantly measure content similarity rather than style similarity.
> 3. Moreover, IP-Adapter and other T2I algorithms also use similar metrics, calculating the **CLIP similarity between the generated image and the style prompt**. Since this is a multimodal task, we believe it is a reasonable approach to adopt metrics that incorporate multimodal inputs.
>
> This metric is indeed not suitable for evaluating performance on color transfer. However, for models aimed at disentangled representations, it is **appropriate for assessing style similarity from a disentanglement perspective**. While some comparison methods exhibit minimal color deviation in visual results, our method achieves better style similarity according to this metric, demonstrating that our approach effectively disentangles the artist's style from the reference image.
>
> - **Why not use Gram Matrix loss for style similarity metric**
>
> The Gram matrix, which computes the correlation between channels of CNN features, primarily measures color and texture similarity. It has predominantly been used as a style loss function in earlier image-to-image style transfer algorithms. In such works, Gram features are rarely employed directly as a metric for evaluating style similarity. Examples include AdaIN, SANet, AdaAttn, Avatar-Net, StyleBank, and others. Although the Gram matrix is used in loss functions to calculate color and texture correlations, it is **less suitable than CLIP-based metrics** for tasks supporting **multimodal inputs**. This metric favors algorithms that define style primarily as color but is not fair for algorithms focusing on the artist’s overall style (such as our method and ArtBank, which also emphasize the artist's overall style).
>
> In contrast, recent studies on **text-to-image(T2I) stylization** methods predominantly adopt multimodal feature similarity in CLIP space as a quantitative metric for assessing style similarity and content consistency. This metric is especially relevant for methods that support **multimodal input**. While we observed that DreamStyler utilizes Gram loss, the majority of **T2I methods**, such as ArtAdapter, ArtBank, DEA-Diff, InstantStyle, IP-Adapter, T2I-Adapter, and StyleTokenizer, **do not use Gram features as an evaluation metric**. Instead, they rely on **CLIP-based metrics** as their primary evaluation criteria.
>
> For the reasons mentioned above, we have not adopted this metric in our study.

---

> > ### Comment · Reviewer_ikfg · 2024-12-03
> >
> > I still concern that the style evaluation metric used in this paper is not very reasonable. Instead of directly calculating the style similarity between the two images, the metric first converts one of the images into a textual form. How can the authors ensure that this process does not lead to any loss or alteration of the style?

---

> > > ### Author Response · Authors · 2024-12-04
> > > **Authors' Reply**
> > >
> > > Thank you for taking the time to review and consider our responses! Regarding your concern about the style similarity metric, we would like to make several clarifications:
> > >
> > > - Metrics that directly compute the similarity between two images, such as Gram feature loss, **can only capture statistical properties** (e.g., texture or local feature correlations) rather than the style of an image (e.g., color schemes, brushstroke techniques). When the content of two images differs significantly, the statistical distribution of low-level features may overshadow the style similarity. For example, **two images in the same style would exhibit large Gram feature differences due to different content composition.**
> > >
> > >    In style transfer tasks involving a reference image and content text, the generated image and the style image often have significant content and compositional differences, making Gram feature loss unsuitable as a metric for style similarity. To our knowledge, only DreamStyler has employed this metric to evaluate style similarity in such tasks. However, in their experimental results, the method that directly copies the entire style image achieved the best score, further demonstrating that this metric is not well-suited as a style evaluation metric.
> > >
> > > - The CLIP-based metric we use would indeed result in some loss of style information during computation; however, it offers some major advantages:
> > >
> > >   - **Text descriptions can directly represent high-level attributes of artistic style**, such as color usage and brushstrokes , which Gram matrices fail to capture.
> > >   - CLIP-based metrics perform calculations in a multimodal feature space, **avoiding interference caused by content and compositional differences** when evaluating style similarity.
> > >
> > >   This metric is also a standard approach adopted in over ten cutting-edge multimodal stylization papers, including ArtAdapter, ArtBank, DEA-Diff, InstantStyle, IP-Adapter, T2I-Adapter, and StyleTokenizer.
> > >
> > > We hope this response addresses your concern.

---

> ### Author Response · Authors · 2024-11-29
> **Additional Quantitative Experiments**
>
> In Appendix 3.2, Table 4, we included quantitative comparisons with SD from Section 5.2.2 (stylized text-to-image generation) and ArtBank from Section 5.2.3 (collection-based stylization tasks). The results show that our method achieves performance metrics **comparable to Stable Diffusion** and **surpasses ArtBank**. In our quantitative experiments across different settings, **SD outperforms all existing methods**.
>
> | Metrics | SD [1] | ArtBank [2] | Ours |
> |---------|--------|-------------|------|
> | SS      | **0.297**  | 0.287       | *0.293* |
> | IQ      | **5.823**  | 5.797       | *5.811* |
> | TA      | **0.321**  | 0.291       | *0.317* |
>
> **Table:** Quantitative comparison with stylized text-to-image generation and collection-based stylization methods.
>
> **References:**
> 1. Rombach et al. (2022). Stable Diffusion.
> 2. Zhang et al. (2024). ArtBank.
>
>
>
> As shown in Figure 5, SD exhibits a more pronounced stylization, which contributes to its superior performance on quantitative metrics. However, as illustrated in Figure 16, SD sometimes also demonstrates **limitations in disentangle content and style**. The generated images include elements not specified in the input images (e.g., row 4, the person in the middle, which is absent from both the style image and the content image).
>
> For the comparison with DALL-E, as it is a **not open-sourced commercial software** intended for testing purposes only, we conducted visual comparisons exclusively (Figure 5). The results reveal that DALL-E often generates repetitive patterns and textures across different style prompts, suggesting that its stylization performance is suboptimal.

---

> ### Author Response · Authors · 2024-11-29
> **Inference Speed**
>
> In Appendix 3.1 Table 3, we compared the inference speed of our method with two structurally similar methods, T2I-Adapter and Dea-Diff in the table below. All three methods utilize the pre-trained SD model as the generative backbone; however, they differ in the conditional modules used to control the diffusion process. The complexity of our method is comparable to that of Dea-Diff, resulting in similar inference times.
>
> | Method       | Ours  | T2I-Adapter | DEA-Diff |
> |--------------|-------|-------------|----------|
> | Time (seconds) | 3.6s  | 4.2s        | 2.2s     |

---

### Official Review · Reviewer_USxT · 2024-11-02

**Soundness:** 3
**Presentation:** 2
**Contribution:** 2
**Rating:** 5
**Confidence:** 4

**Summary:**

This paper collects a new dataset WikiStyle+ and proposes to learn decoupled content-style representations with conventional VLM pre-training losses, which are injected into the diffusion models via cross-attention layers for stylized image generation.

**Strengths:**

1. A new dataset with multimodal annotations of both style and content is collected.

2. Conventional VLM pre-training losses are adopted in the scenario of diffusion model based stylized image generation.

**Weaknesses:**

1.The qualitative results are not convincing. For example, DeaDiff and InstantStyle  better follow the input styles than the proposed method in Fig.1 and Fig.5 respectively.

2.The authors should discuss the differences between WikiStyle+ and the existing WikiArt dataset [A] (e.g., image data overlap). Moreover, the authors do not clarify that whether the dataset will be released in the future.

[A] Recognizing image style.

3.The pipeline in Fig.4 is not consistent with the descriptions in Sec.4.2. For example, (L308-310) the style embeddings are expected to be injected into the diffusion model via multi-step cross-attention layers but in Fig.4, the style embeddigns are injected into the midlle block only and the time embedding is missing as input to the multi-step cross-attetion layers; (L312-313) the content embeddings are expected to be concatenated with the text embeddings from the text encoder but the content embeddings are fed into the cross-attention layers in Fig.4.

4.The ablation study in Sec.5.4 is not sufficient for the missing discussions about the effect of each item in Eq.(1).

**Questions:**

1.How many images in WikiStyle+ overlap with WikiArt dataset? Will the dataset be released?

2.How does each item in Eq.(1) affect the final results?

3.Dose the multi-step cross-attention layers accept the modulation from the time step?

--------Response on Dec 4th----------

Thanks the authors for the further response after the rebuttal period. Two main concerns are still not addressed:

1.The authors claim that “The single reference image is not the only color source for the generated results; the model also utilized color usage patterns embedded in the pre-trained dataset. ” But OOD evaluation is not included in the paper (which I have pointed out during the rebuttal period and the related details are not clarified in the paper as well), where the reference style/artist is not included in the pre-training dataset.

2.The authors claim that “The style of an image should be reflected in the artist's color usage habits rather than the specific colors present in that particular image. These habits encompass commonly used tones (e.g., bright, soft, or dark), color schemes (e.g., complementary, analogous, or monochromatic), as well as attributes like saturation and contrast.” However, the adopted style similarity metric in this paper lacks persuasiveness for comparions among all the competing methods. Also, the metric subjective preference should be disentangled into different aspects (e.g., tones, color schemes, etc).

---

> ### Author Response · Authors · 2024-11-28
> **Qualitative performance analysis with DeaDiff and InstantStyle comparison**
>
> We understand that the perception of style similarity can be subjective, which is likely the main reason behind the reviewer’s impression that our generated results lack style similarity. However, our work adheres to the definition of style in art history, viewing it as the artist’s overall expressive techniques rather than merely replicating specific iconic textures or colors from an individual painting.
>
> Here, we would like to reference the comments made by two other reviewers (fLvE and gsAG). Our method “**captures broader brushstroke techniques of the artist rather than simply replicating signature swirling patterns**” and **“achieved more fine-grained control in generation and yielding excellent results (**fLvE**)”**.
>
> In Figure 1, the results generated by DeaDiff exhibit content leakage from the reference image, as the background of “Starry Night” appears consistently across different themes. We view this not as a follow of style but rather a replication of fixed patterns resulting from content leakage. Similarly, in Figure 4(original 5), InstantStyle also demonstrates content leakage. For example, in the first row, the generated image includes elements such as wheat fields and trees from the reference image, despite these not being mentioned in the content prompt. In contrast, our model explicitly disentangles the content and style of the reference image at the feature level, ensuring that the generated image's content is solely guided by the input content prompt, proving that our approach minimizes content leakage from the reference image.
>
> In Appendix Figure 11, we provide a detailed comparison with InstantStyle and DEA-Diff, including the artist's original painting in the last column to offer a more intuitive demonstration of how our results more closely follow the artist's overall style. From the experimental results, **InstantStyle** focuses on **replicating the colors and content of the reference image**. DEA-Diff, on the other hand, faithfully reflects the content prompt but produces brushstrokes that deviate significantly from the reference image's style, **resembling comic illustrations rather than oil paintings**. In contrast, our method emphasizes imitating the artist's overall style, delivering more stylistically accurate results. This is because **DEA-Diff and InstantStyle define style differently from our approach**. They focus on imitating the color and texture, whereas our method aims to replicate the artist’s overall painting habits.
>
> Under a unified style, our approach **reflects different content in varied yet stylistically consistent ways**. From Figure 11, regardless of the input content, DEA-Diff and InstantStyle consistently produce the **same color palette and textures**, which **do not align with the artist’s actual painting habits**. On the contrary, our generated results incorporate similar colors and distinctive swirl patterns from Starry Night only when the input content prompt explicitly includes "starry night." Additionally, when the content prompt specifies a "city skyline," our results accurately depict urban scenes, while InstantStyle continues to replicate the rural content from the reference image. Meanwhile, DEA-Diff produces results that resemble anime rather than an oil painting. These observations demonstrate that our method effectively disentangles style and content, producing outputs that align more closely with the artist’s painting habit and overall style.
>
> In Appendix, Figure 15 further demonstrates the superiority of our disentanglement design. DeaDiff exhibits several failure cases in disentangle content information from images. For example, under the "pencil" style, the color of the mountains in the content image is incorrectly treated as style information and transferred to the generated result. With "Samuel Peploe" style, DeaDiff struggles to produce a stylized output, instead following the photographic style from the content image. These issues arise because Dea-Diff lacks the explicit disentanglement design proposed in our method. Consequently, it often reproduces the style of the content image or the content of the style image in the generated results.  In contrast, our method effectively avoided content or style leakage in the generated images while  capturing the artist's style.

---

> ### Author Response · Authors · 2024-11-28
> **Difference between WikiStyle+ and dataset from [1] & open source**
>
> Both our proposed WikiStyle+ dataset and the dataset in [1] are based on the publicly available WikiArt website, resulting in some overlap in images.  WikiStyle+ is more than an extension version of [1]—it is designed to address the challenge of disentangling content and style, a foundational problem in artistic stylization.  The primary difference between WikiStyle+ and the dataset from [1] lies in their **scale**, **diversity**, and **data structure**:
>
> 1. **Scale and Diversity**:
> - WikiStyle+ contains over **146k images** spanning **209 styles**, while the dataset from [1] includes **85k images** covering **25 styles**. This broadens  WikiStyle+’s applicability for various downstream tasks.
> 2. **Data Structure**:
> - WikiStyle+ introduces **artistic image–style description–content description triplets**, making it the **first dataset of its kind**.  This structure not only supports disentanglement but also enables multimodal tasks such as style transfer, retrieval, and generation.
>
> We commit to **open-sourcing** the dataset as well as source code upon acceptance, as we believe it holds significant value for advancing research in content-style disentanglement—a critical challenge in artistic stylization.

---

> ### Author Response · Authors · 2024-11-28
> **Ablation study for each item in Eq.(1)**
>
> In Eq.(1), **Image-Text Contrastive Loss (ITC)** primarily handles the tasks of alignment and disentanglement in the network, while the **Image-Text Matching Loss (ITM)** and **Image-grounded Text Generation Loss (ITG)** provide auxiliary support for modality alignment based on ITC. To evaluate the contribution of each loss function and their impact on final results, we conducted an ablation study in Appendix 3.3, where four different configurations of the model are compared:
>
> 1. Training **with only Image-Text Contrastive Loss (ITC)**. ITC is the core loss function in the first stage, enables the model to disentangle style and content more effectively by ensuring that features corresponding to style and content are aligned with their respective textual descriptions.
> When the CSDN is trained exclusively with ITC, the overall content structure is preserved. When the model is trained solely with ITC, the content alignment score achieves the highest value among all configurations. This indicates that $\mathcal{L}_{\mathit{itc}}$ as a contrastive loss, is highly effective in aligning the content features with their corresponding textual descriptions. However, the generated results lack intricate brushstroke details, and fail to capture the characteristic yellowish tone of traditional oil paintings, as illustrated in **Fig. 10(a)**. The lack of fine-grained alignment with styles is corroborated by the decreased SS metrics in **Table 5**.
>
> 2. Training **without Image-grounded Text Generation Loss (ITG)**: ITG trains the model to generate coherent style and content descriptions for a given image by predicting the next word or sentence based on the image and context. ITG ensures that the style and content information extracted from an image is not only disentangled but also interpretable and coherent, which enhances the overall textual-visual understanding of the model. As shown in **Fig.10 (b)**, the absence of ITG affects the coherence of the style and content descriptions extracted from the image. For example, in the "A bus" prompt, the generated bus elements are inconsistent. Also, stylistic elements such as brushstrokes are less pronounced, leading to the drop of SS and TA in quantitative results.
>
> 3. Training **without Image-Text Matching Loss (ITM)**: ITM operates as a binary classification task, predicting whether an image-text pair is a positive or negative match. This enables the model to focus on fine-grained correspondence between images and text, such as specific objects or elements mentioned in the prompts. When ITM is removed, the generated images show a loss of detail in both style and content alignment. For instance, in **Fig. 10 (c)**, the ``house'' element explicitly mentioned in the content prompt is missing in the generated image. This indicates that, without ITM, the model’s capacity to maintain fine-grained alignment is compromised, resulting in less coherent and contextually relevant outputs.
>
> 4. **Full-loss setting**: When all three loss functions are used together, the SS metrics achieve the highest scores. From **Fig. 10 (d)**, the generated images align closely with both style and content prompts, showcasing strong disentanglement and alignment. The style is faithfully preserved, while the content, such as the "house" or "bus," is accurately represented in the generated outputs. This demonstrates the complementary nature of the three loss functions in ensuring both disentanglement and multimodal coherence. While ITC prioritizes content alignment, the inclusion of ITM and ITG shifts the focus toward achieving a better balance, where both content and style are accurately disentangled and aligned with their textual descriptions.
>
> In conclusion:
>
> - **ITC** ensures effective alignment of visual and textual features.
> - **ITG** ensures that the style and content information extracted from an image is not only disentangled but also interpretable and coherent. Without ITG, the generated images may lose stylistic fidelity or fail to fully capture content prompts, as the model would lack a strong textual grounding during training.
> - **ITM** enhances fine-grained alignment between textual descriptions and corresponding visual elements, enabling the model to accurately reflect details in the generated images. Without ITM, the relevance of generated images to the input prompts degrades, with style and content features becoming less accurately matched to their textual descriptions.

---

> ### Comment · Reviewer_USxT · 2024-11-28
>
> Thanks for the efforts for resolving my concerns about ablation study, model details and dataset comparisons. My main concern is still not addressed:
> 1) Generally, the overall painting habits of an artist include both the strokes and colors. With only one single out-of-domain reference image as input, the stylized image should be aligned with both of these terms. In this task, the content leakage from the reference image is hard to define. It is inappropriate to regard all the semantic objects with unusual colors as content leakage in art work generation.
>
> 2) The authors claim that the proposed method can better retain the artist’s actual painting habits than existing methods. However, overall color deviation from the reference images is also observed (e.g., in Fig. 4, 6, 11, 14 and 16).
>
> Considering the revised paper and detailed responses from the authors, I change my rating from 3 to 5.

---

> ### Author Response · Authors · 2024-11-28
> **Framework and description clarification**
>
> - We would like to clarify that the term **"multi-step"** refers to the **multiple denoising steps** during the training process of the diffusion model, rather than indicating that the style embeddings are injected into multiple blocks of the U-Net. In our implementation, the style embeddings are indeed **only injected into the** **middle block** of the U-Net through cross-attention at every denoising step across the diffusion process. To clarify this point, we have revised the original text in L311, changing **"we inject the style embeddings only into the intermediate layer of the denoising process"** to **"we inject the style embeddings only into the middle block of U-Net"** (L312 in the revised PDF).
> - For the content text, we first extract text features using the original text encoder from SD, then concatenate them with the disentangled content embeddings $e_c$ extracted by CSDN before injecting them into the diffusion process. Previously, we simplified this step in the flowchart, but for better clarity, we have explicitly **added this step to Figure 3 (formerly Figure 4)** in the revised PDF.
>
> - Dose the multi-step cross-attention layers accept the modulation from the time step?
>
>   * The multi-step cross-attention layers (MCL) in our model do not directly accept modulation from the time step embeddings. Instead, time steps are incorporated into the U-Net's processing pipeline through the standard embedding mechanism of Stable Diffusion (SD). This embedding is injected into each layer of the U-Net to modulate its feature processing based on the current diffusion step. In our implementation, MCL specifically focuses on incorporating the disentangled style and content embeddings into the denoising process via cross-attention. While the time step information indirectly influences the MCL through the modulated features of the U-Net, it is not explicitly part of the attention mechanism in the MCL. We **omitted the detailed depiction of the time step embedding process** in the framework for simplicity, as it **follows the standard practice of SD and is orthogonal to the design of our MCL**.

---

> ### Author Response · Authors · 2024-12-04
> **Authors' Reply**
>
> Thank you for taking the time to review and consider our responses!
>
> - **The style of an image should be reflected in the artist's color usage habits rather than the specific colors present in that particular image.** These habits encompass commonly used tones (e.g., bright, soft, or dark), color schemes (e.g., complementary, analogous, or monochromatic), as well as attributes like saturation and contrast. If style were defined solely by the specific colors in an image, then *by that logic, each of Van Gogh's works with a different color palette would represent a different style.*
>
> - **The single reference image is not the only color source for the generated results; the model also utilized color usage patterns embedded in the pre-trained dataset.** In pretraining, the model learns from large-scale artworks and clusters similar color usage patterns (e.g., artists who paint blue skies also tend to depict sunflowers in gold). In the inference stage, the model extracts color habits from the reference image and combines this with the knowledge acquired during pretraining. As a result, the model does not simply transfer colors from the reference image but infers suitable colors for different content, ensuring better stylistic fidelity. *Otherwise, when the content prompt is "sunflowers" and the reference image is Van Gogh's Starry Night, would generating "blue sunflowers sparkling with stars" truly reflect Van Gogh’s style?*
>
> These clarifications may be insufficient in the manuscript, leading to potential misunderstandings. We have revised the introduction section based on the response above. We hope this response addresses the reviewer's concern.

---

### Author Response · Authors · 2024-11-28
**Summary of Author rebuttal**

We sincerely thank the reviewers for reviewing our submission. We are encouraged with positive feedbacks on the excellent results and better **style resemblence** (**fLvE**, **gsAG**), **extensive experiments** (**ikfg**, **fLvE**), well-motivated method and dataset(**ikfg, fLvE, gsAG**) and **well-written and presentation** (**fLvE**). We deeply appreciate the reviewers' valuable feedback and constructive suggestions, which have been instrumental in improving the quality of our manuscript.

In the revised PDF, we included supplementary experimental results in the **appendix** to support our rebuttal. Additionally, we highlighted the revised sections in **red** for better clarity.

- A.3.1, Table 3: Inference speed. (Reviewer **ikfg**)
- A.3.2, Table 4: Quantitative comparison with stylized text-to-image generation and collection-based stylization methods. (Reviewer **ikfg**)
- A.3.3, Figure 10 and Table 5 : Ablation Study for each item in Eq(1) (Reviewer **USxT,fLvE**)
- A.3.4, Figure 11 : Style resemblance comparison with Dea-Diff and InstantStyle (Reviewer **USxT, ikfg**)
- A.3.5, Figure 12: Impact of Content and Style Description Detail Levels on Disentanglement Performance（Reviewer **gsAG**）
- A.3.7, Figure 13: Visual comparison with StyleDrop. (Reviewer **ikfg**, **gsAG**)
- A.3.8, Figure 14: Visual comparison with DreamStyler. (Reviewer **ikfg**, **gsAG**)

We respond to each reviewer below to address their concerns. We hope our responses adequately resolve the reviewers' concerns, but please take a look and let us know if further clarification or discussion is needed.

---

> ### Author Response · Authors · 2024-12-04
> **Summary of Second Round Rebuttal**
>
> We sincerely thank all the reviewers for taking the time to review our rebuttal and thoughtfully consider our responses!
>
> We address each reviewer’s second-round concerns below. We hope our responses sufficiently resolve these issues and invite you to review them.
>
> - Color deviation from single reference image (Reviewer **USxT**)
>
> - Style similarity metric (Reviewer **ikfg**)
>
> - Learn style from single reference image (Reviewer **ikfg**)
>
> - Experimental setting in comparison with DreamStyler (Reviewer **gsAG**)

---

### Meta-Review · Area_Chair_mJje · 2024-12-21

**Metareview:**

This paper studies controllable artistic image stylization and generation. The Area Chair has read all reviews and authors' responses. The overall rating is below the acceptance threshold.

The reviewers acknowledged several strengths of the paper, including the studied problem being important and the method being well-motivated, the introduced dataset wikistyle+ with multi-modal annotation being valuable, and the experiments being extensive and the performance surpassing the baseline methods.

Yet, the reviewers also raised some significant concerns, including some qualitative results being not fully convincing, the style evaluation metric used in this paper being not rigorous, and several concerns regarding specific experimental setups and missing comparison methods. There are also concerns regarding its novelty being incremental and writing issues (unclear description, ambiguous symbols, and typos). The authors' rebuttal provided helpful explanations and addressed some of the concerns. While reviewers acknowledged these improvements and adjusted their ratings accordingly, some of the primary issues remained unresolved. As a result, the overall rating still falls short of the acceptance threshold.

**Additional Comments On Reviewer Discussion:**

After rebuttal, the reviewers acknowledged that some of their concerns were addressed and raised their ratings. Yet, the reviewers' main concerns are not fully addressed and the overall rating doesn't reach the acceptance threshold.

---

### Decision · Program_Chairs · 2025-01-22

Reject